# Cerebellar associative sensory learning defects in five mouse autism models

Alexander D Kloth[1†], Aleksandra Badura[1], Amy Li[1], Adriana Cherskov[1], Sara G Connolly[1], Andrea Giovannucci[1], M Ali Bangash[2], Giorgio Grasselli[3], Olga Peñagarikano[4,5‡], Claire Piochon[3], Peter T Tsai[6§], Daniel H Geschwind[4,5], Christian Hansel[3], Mustafa Sahin[6], Toru Takumi[7], Paul F Worley[2], Samuel S-H Wang[1]*

[1]Department of Molecular Biology and Princeton Neuroscience Institute, Princeton University, Princeton, United States; [2]Department of Neuroscience, Johns Hopkins University School of Medicine, Baltimore, United States; [3]Department of Neurobiology, University of Chicago, Chicago, United States; [4]Department of Neurology, David Geffen School of Medicine, University of California, Los Angeles, Los Angeles, United States; [5]Center for Autism Research, Semel Institute, David Geffen School of Medicine, University of California, Los Angeles, Los Angeles, United States; [6]The F.M. Kirby Neurobiology Center, Department of Neurology, Children's Hospital Boston, Harvard Medical School, Boston, United States; [7]RIKEN Brain Science Institute, Wako, Japan

*For correspondence: sswang@ princeton.edu

Present address: [†]Department of Cell Biology and Physiology, UNC Neuroscience Center, University of North Carolina at Chapel Hill, Chapel Hill, United States; [‡]Department of Pharmacology, School of Medicine, University of the Basque Country, Leica, Spain; [§]Department of Neurology and Neurotherapeutics, University of Texas Southwestern Medical Center, Dallas, United States

Competing interests: The authors declare that no competing interests exist.

**Abstract** Sensory integration difficulties have been reported in autism, but their underlying brain-circuit mechanisms are underexplored. Using five autism-related mouse models, *Shank3*+/ΔC, *Mecp2*[R308/Y], *Cntnap2*−/−, L7-Tsc1 (*L7/Pcp2*[Cre]*::Tsc1*[flox/+]), and patDp(15q11-13)/+, we report specific perturbations in delay eyeblink conditioning, a form of associative sensory learning requiring cerebellar plasticity. By distinguishing perturbations in the probability and characteristics of learned responses, we found that probability was reduced in *Cntnap2*−/−, patDp(15q11-13)/+, and *L7/Pcp2*[Cre]*::Tsc1*[flox/+], which are associated with Purkinje-cell/deep-nuclear gene expression, along with *Shank3*+/ΔC. Amplitudes were smaller in *L7/Pcp2*[Cre]*::Tsc1*[flox/+] as well as *Shank3*+/ΔC and *Mecp2*[R308/Y], which are associated with granule cell pathway expression. *Shank3*+/ΔC and *Mecp2*[R308/Y] also showed aberrant response timing and reduced Purkinje-cell dendritic spine density. Overall, our observations are potentially accounted for by defects in instructed learning in the olivocerebellar loop and response representation in the granule cell pathway. Our findings indicate that defects in associative temporal binding of sensory events are widespread in autism mouse models.

## Introduction

In autism spectrum disorder (ASD; hereafter referred to as autism), atypical sensory processing is widely reported starting in infancy (*Leekam et al., 2007*; *Markram and Markram, 2010*; *Dinstein et al., 2012*). In addition to early-life abnormal processing of single sensory modalities (*Leekam et al., 2007*), more complex deficits become apparent as early as 2 years of age, a time when autistic children attend poorly to natural combinations of spoken stimuli and natural visual motion (*Klin et al., 2009*), a circumstance that calls upon the ability to integrate, from moment to moment, information from two sensory modalities, hearing and vision. Abnormalities of sensory responsiveness are strongly correlated with severity of social phenotypes in high-functioning autism patients (*Hilton et al., 2010*). Taken together, these observations suggest that abnormal processing of multiple sensory modalities on subsecond time scales might impede the acquisition of cognitive and affective capacities that are affected in autism.

**eLife digest** On a windy day, hearing the sound of wind makes many individuals squint in anticipation in order to protect their eyes. Linking two sensations that arrive within a split second of one another, such as sound and the feeling of wind, is a type of learning that requires the cerebellum, a region found at the base of the brain. When done in a laboratory setting, this particular form of learning has been dubbed eyeblink conditioning.

Individuals with autism tend to have difficulties with appropriate matching of different senses. For example, they have trouble identifying a video that goes with a spoken soundtrack. They also do not learn eyeblink conditioning the same way that other individuals do. However, it is not known which circuits in the brain are responsible for their difficulty. Kloth et al. now investigate this issue by asking whether versions of genes that increase the risk of autism in humans also disrupt eyeblink conditioning in mice. They tested five types of mouse model, each with a different genetic mutation that has previously been linked to autism. All five of these mutations cause defects in different cell types of the cerebellum, and all mice have abnormal social and habitual behaviors, similar to autistic people.

The tests involved shining a bright light at the mice, which was followed, a split second later, by a puff of air that always causes the mice to blink. After this had occurred dozens of times, the mice started to blink earlier, as soon as the light appeared, in anticipation of the puff of air. To test whether the mice had successfully learned to respond to just the bright light, the light was also occasionally flashed without a puff of air.

Kloth et al. found that the mice generally performed poorly in eyeblink conditioning, although in different ways depending on which cell types of the cerebellum were affected by the genetic mutations. Some mice blinked too soon or too late after the light appeared; others blinked weakly or less frequently; and some did not blink at all. This suggests that autism can affect the processing of sensory information in the cerebellum in different ways.

This work is important because it demonstrates that a form of split-second multisensory learning is generally disrupted by autism genes. If defects in cerebellar learning are present early in life, they could keep autistic children from learning about the world around them, and drive their developing brains off track. Hundreds of autism genes have been found. Linking these genes to a single brain region identifies the cerebellum as an important anatomical target for future diagnosis and intervention.

Abnormal sensory processing in autism is likely to arise in part from genetic mutations and variants that predispose for neural circuit dysfunction. To investigate the ability to associate two near-simultaneous sensory inputs, we used delay eyeblink conditioning, a form of learning that is found in multiple mammalian species (*McCormick and Thompson, 1984*; *Ivarsson and Hesslow, 1994*; *Boele et al., 2010*; *Heiney et al., 2014*). Persons with autism show alterations to delay eyeblink conditioning (*Sears et al., 1994*; *Oristaglio et al., 2013*). Delay eyeblink conditioning depends on plasticity in the cerebellum, a common site of anatomical deviation in patients with autism, and cerebellar gross and cellular malformation are common features of autistic brains (*Wang et al., 2014*). These factors led us to search for aberrations in the quantitative parameters of delay eyeblink conditioning.

Autism is among the most heritable of neuropsychiatric disorders (*Gaugler et al., 2014*), and hundreds of autism risk loci have been identified (*Abrahams and Geschwind, 2008*; *Devlin and Scherer, 2012*; *Stein et al., 2013*). We examined five mouse models that both recapitulate mutations that occur in human idiopathic and syndromic autisms and display phenotypes reminiscent of human autism (*Abrahams and Geschwind, 2008*; *Banerjee-Basu and Packer, 2010*; *Abrahams et al., 2013*; http://gene.sfari.org). Four of the models incorporate global mutations with strong expression in the cerebellum: *Shank3+/ΔC*, the C-terminal deletion model of *Shank3* associated with Phelan-McDermid syndrome (*Kouser et al., 2011*, *2013*); *Mecp2$^{R308/Y}$*, a mild truncation model of *Mecp2* associated with Rett syndrome (*Ben-Shachar et al., 2009*; *Shahbazian et al., 2002a*; *Moretti et al., 2006*; *De Filippis et al., 2010*); *Cntnap2−/−*, a knockout of *Cntnap2* associated with cortical dysplasia-focal epilepsy syndrome (*Peñagarikano et al., 2011*); and patDp/+, a mouse model of the 15q(11–13) duplication syndrome closely linked to autism (*Nakatani et al., 2009*; *Tamada et al., 2010*;

*Piochon et al., 2014*). A fifth model, a knockout of the tuberous sclerosis protein L7-Tsc1 (*L7/Pcp2$^{Cre}$::Tsc1$^{flox/+}$* and *L7/Pcp2$^{Cre}$::Tsc1$^{flox/flox}$*), specifically affects cerebellar Purkinje cells (PCs) (*Tsai et al., 2012*).

Because different circuit defects might have differential effects on the properties of eyeblink conditioning, we analyzed learning deficits quantitatively in terms of two major features of learning: the probability of generating a response, reflecting the learning process itself; and the magnitude and timing of individual responses, reflecting the neural representation of the learned response.

## Results

All five mouse models examined in this study have previously shown face validity for autism (*Silverman et al., 2010*), with alterations in social behavior, ultrasonic vocalization, and repetitive behaviors. Some, but not all, of these models show disruptions of gross motor function. *Cntnap2−/−* mice and patDp/+ mice show enhanced performance on a gross motor task, the accelerating rotarod (*Nakatani et al., 2009*; *Peñagarikano et al., 2011*); but the other three mouse models do not (*Shahbazian et al., 2002a*; *Kouser et al., 2011*; *Tsai et al., 2012*). In addition, patDp/+ has been tested and shown to have alterations in gait (*Piochon et al., 2014*). We surmised that a more refined assay might reveal cerebellum-specific functional disruptions.

We subjected head-fixed mice to delay eyeblink conditioning (*Figure 1A*; *Arlt et al., 2010*; *Heiney et al., 2014*; *Piochon et al., 2014*). Over the course of training with a light-flash conditioned stimulus (CS; ultraviolet LED, 280 ms) and a co-terminating corneal-airpuff unconditioned stimulus (US; 30 ms), a conditioned response (CR) developed with a gradually rising time course that peaked at the time of the US onset (*Figure 1C*). During each training session (220 trials), a small number (10% CS-only trials) of unpaired CS (i.e., no US) trials were used to characterize the complete CR time course, including the onset time, the rise time, and the peak time (*Figure 1B*). Finally, to probe savings, an aspect of eyeblink conditioning that depends in part on the deep cerebellar nuclei (DCN), after the 12-day initial training period we tested extinction and reacquisition (*Figure 1D*; *Medina et al., 2001*; *Robleto et al., 2004*; *Ohyama et al., 2006*). Extinction consisted of 110 trials of CS-only trials and 110 trials of US-only trials over four daily sessions and led to the near-disappearance of the CR (*Figure 1D*). Three sessions of reacquisition (identical to acquisition) resulted in a rapid return of the CR (*Figure 1D*).

In order to separate the learning process from the learned response, we analyzed session-by-session sets of responses to distinguish the *probability* of generating a CS-evoked eyelid deflection from the *amplitude* of the eyelid deflection on trials when a response occurred (*Garcia et al., 2003*). To estimate the probability of generating a response, we used the overall distribution of eyelid movement amplitudes (*Figure 2*; *Kehoe et al., 2008*, *2009*). First, we computed frequency histograms of the normalized eyelid movement amplitudes occurring between 100 ms and 250 ms after the CS onset (*Figure 2A*; for representative data, see *Figure 2B*, top). A peak in the histogram consistently occurred within the zero-amplitude bin (peak at amplitude of 0.006 ± 0.001, within the bin from −0.0125 to 0.0125), representing failure to respond to the stimulus with either closing or opening of the eyelid. We reflected the histogram of negative-amplitude responses across the zero axis and took the integral of the resulting distribution as the failure rate (*Figure 2A*, light gray). Response *probability* was defined as one minus the failure rate. The average response *amplitude* was calculated as the center of mass of the remaining distribution after subtracting the failure histogram (*Figure 2A*, black). Finally, in addition to probability and amplitude, we calculated three timing parameters of the average learned response: latency to onset of the blink, latency to peak, and rise time.

To test whether variation in wild-type littermates might be a source of apparent differences in autism-model mouse eyeblink conditioning, we compared learning and timing parameters across all wild-type control groups (*Figure 1—source data 1*). We found no significant difference among wild-type cohorts in any learning parameter. In addition, we did not find statistically significant differences in time course of extinction or reacquisition. Because of the residual possibility of undetected variations (e.g., arising from a mixed background for the L7-Tsc1 cohort vs a C57B/6J background for all other groups) and changes in environmental conditions over the period of this study, we used wild-type littermates as a basis for comparison for each autism mutant group (*Crawley, 2008*).

### Defects of CR probability

Three mouse models showed deficits in the response probability during training. In L7-Tsc1 mice (*Figure 3A*), heterozygous mutant mice (*L7/Pcp2$^{Cre}$::Tsc1$^{flox/+}$* or HET, n = 18) reached a response

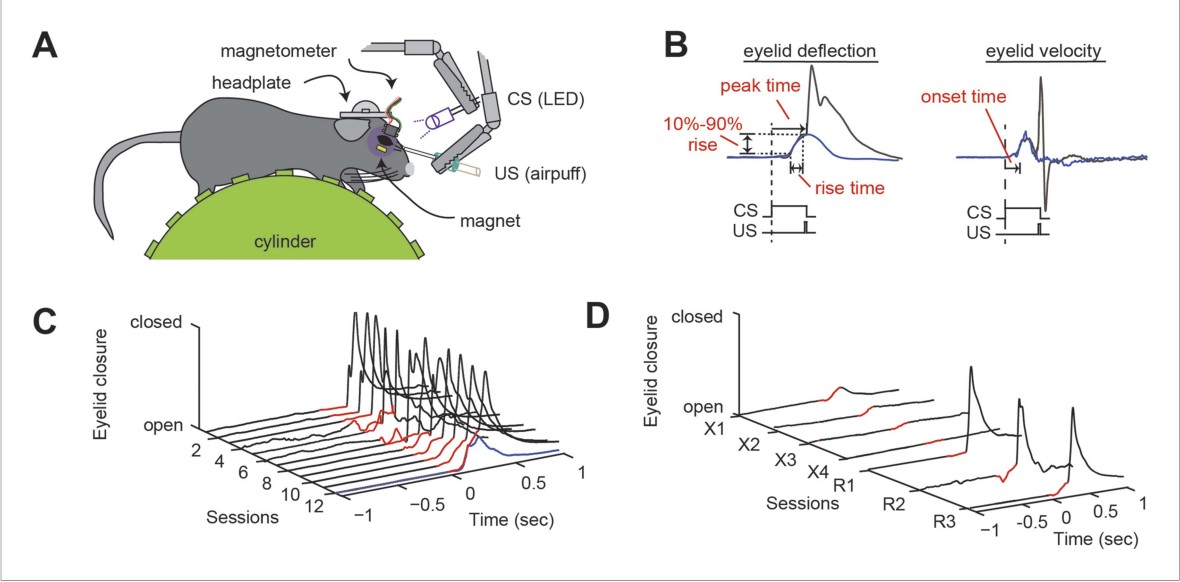

**Figure 1.** Delay eyeblink conditioning in head-fixed mice. (**A**) Experimental setup. A mouse with an implanted headplate is head-fixed above a stationary foam cylinder, allowing the mouse to locomote freely. Eyeblink conditioning is carried out by delivering an aversive unconditioned stimulus (US, airpuff) that coterminates with a conditioned stimulus (CS, LED) to the same eye. Eyelid deflection is measured using induced current from a small magnet affixed to the eyelid. (**B**) When delivered to a trained animal, the co-terminating CS and US produce an anticipatory eyelid deflection (the conditioned response, CR) followed by a reflex blink evoked by the US. When the CS is delivered alone (blue trace), a bell-shaped CR is produced that peaks at the expected time of the US. The onset time is the time from the onset of the CS to a change in concavity of the eyeblink. The rise time is the amount of time between 10% and 90% of the maximum amplitude of the CR (10–90% rise). (**C**) Over twelve training sessions, the CR (portion of trace preceding US, indicated in red) develops in response to the US-CS pairing. One CS-alone response is shown as a blue trace. (**D**) Over four sessions of extinction training, the CR (red) disappears. After three sessions of reacquisition training, the CR (red) returns. *Figure 1—source data 1* provides a wild-type benchmark for the eyeblink parameters described here, along with a statistical analysis of possible difference among wild-type cohorts (p > 0.05 in all instances).

The following source data is available for figure 1:

**Source data 1**. Wild-type values for eyeblink conditioning parameters.

probability of 32.0 ± 4.3%, significantly lower than the 51.5 ± 3.5% level reached in control littermates (n = 16) (last four training sessions; unpaired two-sample t-test, p = 0.01; effect size, Cohen's d' = 1.21). Furthermore, homozygous mutant mice (*L7/Pcp2^{Cre}::Tsc1^{flox/+}* or MUT, n = 5) completely failed to acquire CRs (1.4% ± 0.7% in *L7/Pcp2^{Cre}::Tsc1^{flox/flox}*, n = 5; one-way analysis of variance test (ANOVA) across all groups, p < 0.0001, F(2,35) = 19.82, with Bonferroni post hoc statistical differences between *L7/Pcp2^{Cre}::Tsc1^{flox/flox}* and wild-type littermates, p = 3 × 10^{-9}, Cohen's d' = 3.01, and *L7/Pcp2^{Cre}::Tsc1^{flox/+}* and wild-type littermates, p = 0.00002, d' = 1.41). Further analysis of L7-Tsc1 mice focused on *L7/Pcp2^{Cre}::Tsc1^{flox/+}* only.

In *Cntnap2* mice (**Figure 3B**), homozygous mutant mice (*Cntnap2–/–*, n = 12) reached a response probability of 35.1% ± 6.2%, significantly lower than the 57.2% ± 2.9% level reached in wild-type littermates (*Cntnap2+/+*, n = 13) (last four training sessions; Bonferroni post hoc test after one-way ANOVA, p = 0.007, d' = 0.96). Notably, *Cntnap2+/–* mice, which show behavioral similarity to *Cntnap2+/+* mice (*Peñagarikano et al., 2011*), were likewise statistically indistinguishable in learning or response amplitude from wild-type mice (n = 14 mice; Bonferroni post hoc tests after one-way ANOVA, p > 0.5).

In *Shank3ΔC* mice (**Figure 3C**), the heterozygous mutant mice (*Shank3+/ΔC*, n = 17) reached a response probability of 55.9% ± 3.7%, lower than the 67.2% ± 2.2% reached in the wild-type littermates (*Shank3+/+*, n = 21) (unpaired two-sample t-test, p = 0.015, d' = 1.10). In all three mouse models, probability deficits were present throughout training (two-way repeated measures ANOVA, main genotype effect; *Cntnap2–/–*: F(1,23) = 7.72, p = 0.01; *L7/Pcp2^{Cre}::Tsc1^{flox/flox}*: F(1,23) = 11.70, p = 0.002; *Shank3+/ΔC*: F(1,25) = 4.59, p = 0.04).

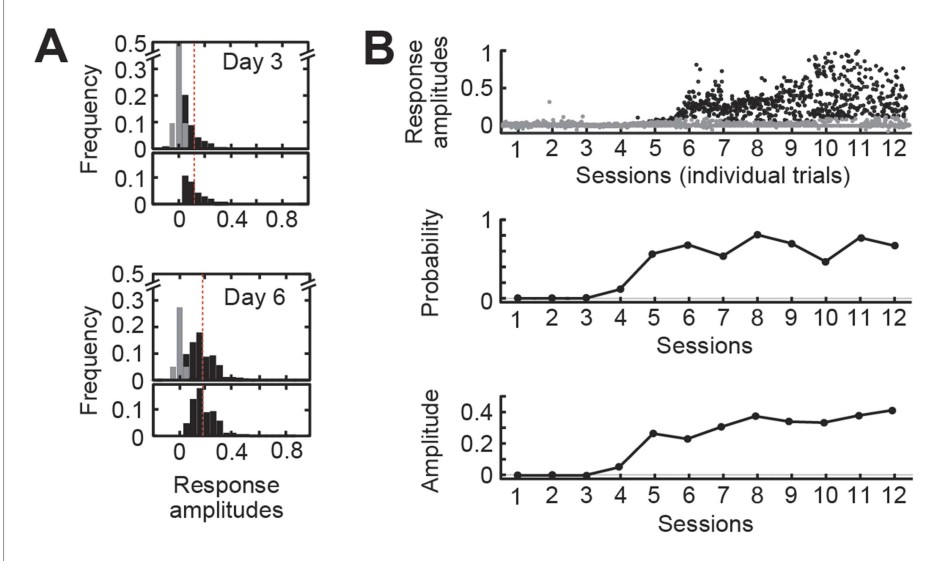

**Figure 2**. Analysis of the full range of detectable responses allows the separation of response probability from response amplitude. (**A**) Response and non-response distributions from days 3 to 6 of training in a single animal. In the top panel for each day, gray bars show the distribution of non-responding trials. In the bottom panel, black bars show the remaining response distribution. The response probability is defined as the area under the response distribution. The average response amplitude is defined as the center of mass of the response distribution. The red line shows the fixed threshold at 0.15. (**B**) Representative data from a single wild-type animal. Top: scatterplot of individual response magnitudes for every trial over 12 sessions of training. Gray dots, individual non-responding trials. Black dots, responding trials. Middle: response probability for each session. Bottom, response amplitude for each session.

One model did not show differences in learning probability or time course: *Mecp2*[R308/Y] heterozygotes (*Figure 3D*; 57.2% ± 2.9% WT vs 57.8% ± 3.6% *Mecp2*[R308/Y], unpaired two-sample t-test, p = 0.9; two-way repeated measures ANOVA: main genotype effect, $F_{(1,22)} = 0.10$, p = 0.7).

We also applied our new analysis technique to a data set previously gathered by our group on the 15q duplication model mice (*Piochon et al., 2014*). We detected a significant difference in response probability that was consistent with previously observed impairment. Throughout acquisition training, response probability in patDp/+ mice (n = 10) was smaller than wild-type littermates (n = 11) (two-way repeated measures ANOVA: main genotype effect, $F_{(1,19)} = 19.84$, p = 0.0003), culminating in a difference at the end of training (34.2% ± 2.9% patDp/+ vs, 49.2% ± 2.6% WT, unpaired two-sample t-test, p = 0.001, d′ = 1.46).

In summary, the five models showed a gradient of defects in probability, ranging from *L7/Pcp2*[Cre]:: *Tsc1*[flox/flox] (no learning) to Mecp2[R308] heterozygotes (intact learning) (*Figure 3E*).

## Defects of CR amplitude

To test whether learned blinks were disrupted, we measured their amplitude normalizing to an unconditioned reflex blink amplitude of 1. After 12 days of acquisition training, three mutant models showed deficits in response amplitude: *L7/Pcp2*[Cre]::*Tsc1*[flox/+], *Shank3*+/ΔC, and *Mecp2*[R308/Y]. *L7/Pcp2*[Cre]::*Tsc1*[flox/+] mice generated smaller-amplitude learned blinks throughout training (two-way repeated measures ANOVA: main genotype effect, $F_{(1,23)} = 7.71$ p = 0.01) that culminated in a difference in amplitude at the end of training (last four training sessions: 0.28 ± 0.03 in *L7/Pcp2*[Cre]:: *Tsc1*[flox/+] vs 0.39 ± 0.05 in littermate controls, unpaired two-sample t-test, p = 0.02, d′ = 0.86) (*Figure 4A*, right). In Shank3ΔC mice (*Figure 4C*), response amplitude was similar to wild-type for most of training (main genotype effect, $F_{(1,24)} = 1.45$, p = 0.2), but culminated in a small reduction by the end of training (0.31 ± 0.02 *Shank3*+/ΔC vs 0.36 ± 0.01 *Shank3*+/+, p = 0.03, d′ = 0.38). *Mecp2*[R308/Y] mice (*Figure 4D*; n = 11) showed consistently smaller learned responses throughout training (two-way repeated measures ANOVA: main genotype effect: $F_{(1,22)} = 12.72$, p = 0.002),

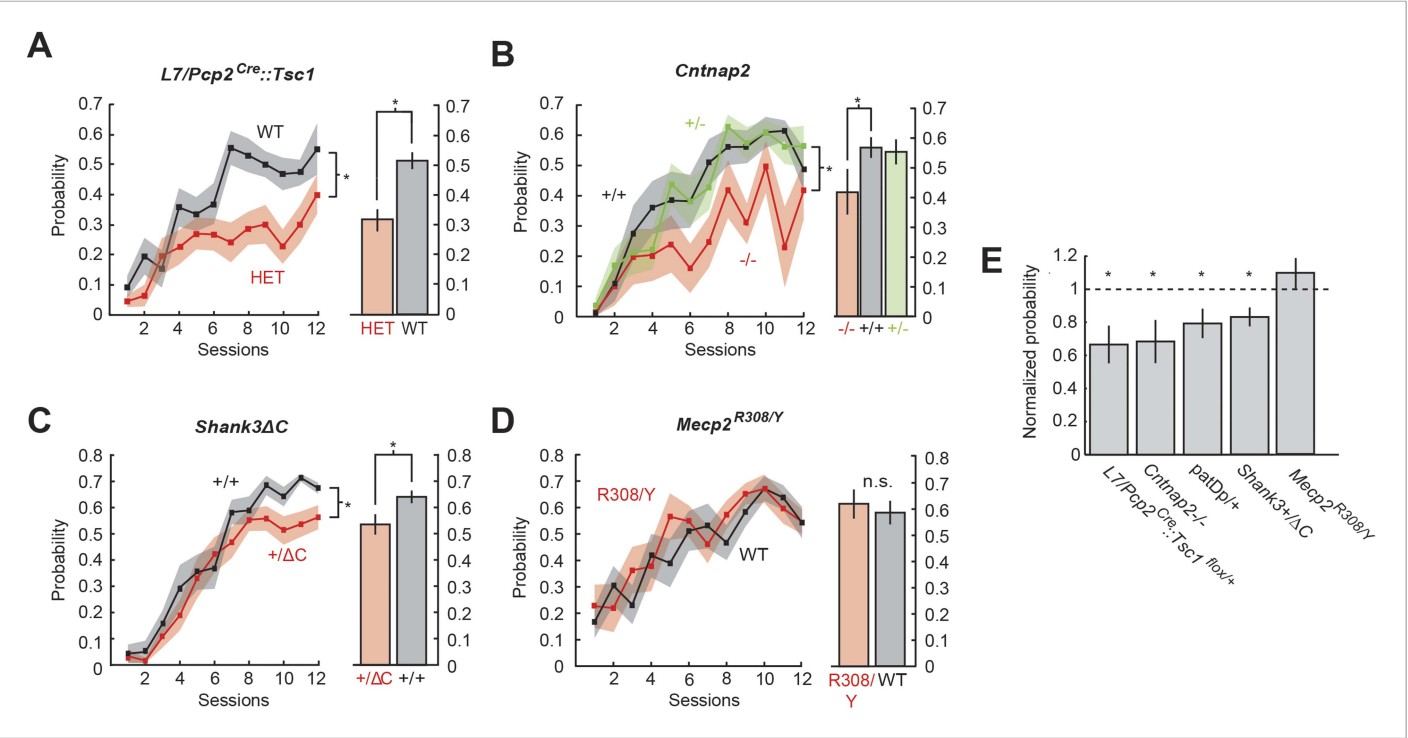

**Figure 3**. Probability defects are present in four mouse models. (**A**) Time course of response probability with acquisition training in L7-Tsc1 model mice. Black: WT. Red: *L7/Pcp2$^{Cre}$::Tsc1$^{flox/+}$*. (**B**) Time course of response probability with acquisition training in *Cntnap2* model mice. Black: *Cntnap2+/+*. Red: *Cntnap2–/–*. Green: *Cntnap2+/–*. (**C**) Time course of response probability with acquisition training in *Shank3ΔC*. Black: *Shank3+/+*. Red: *Shank3+/ΔC*. (**D**) Time course of response probability with acquisition training in *Mecp2$^{R308}$*. Black: WT. Red: *Mecp2$^{R308/Y}$*. In panels (**A**) through (**D**), bar plots indicate response probability averaged over the last four training sessions. (**E**) Probability deficits across all groups. Dashed line: normalized wild-type littermate level. In all panels, shading and error bars indicate SEM, and * indicates p < 0.05. n ≥ 10 mice for each group. *Figure 3—figure supplement 1* shows response probability in each group of animals during extinction and reacquisition.

The following figure supplement is available for figure 3:

**Figure supplement 1**. Extinction and reacquisition.

culminating in a difference in amplitude at the end of training (last four training sessions, 0.28 ± 0.02 *Mecp2$^{R308/Y}$* in vs 0.44 ± 0.04 WT, unpaired two-sample t-test, p = 0.002, d′ = 1.11). CRs in *Mecp2$^{R308/Y}$* mice also reached maximum amplitude much earlier in the training period (*Figure 4D*).

We did not observe statistically significant differences in response amplitude or its development in *Cntnap2* mice (two-way repeated measures ANOVA: main genotype effect, F(2,32) = 0.15, p = 0.85; 0.32 ± 0.03 *Cntnap2–/–* vs 0.34 ± 0.02 *Cntnap2+/+*, Bonferroni post hoc test after one-way ANOVA, p = 0.82) (*Figure 4B*, right) or in 15q duplication mice (two-way repeated measures ANOVA: main genotype effect, F(1,19) = 1.81, p = 0.2), including at the end of training (last four training sessions 0.31 ± 0.02 WT vs 0.27 ± 0.05 patDp/+, unpaired two-sample t-test, p = 0.4; also see *Piochon et al., 2014*). In summary, defects in blink amplitude ranged from large effects exceeding 1 standard deviation (*Mecp2$^{R308/Y}$*) to no statistically detectable difference (*Cntnap2–/–* and patDp/+; *Figure 4E*).

## Normal extinction and reacquisition of CRs

We asked whether CR extinction and savings, two learning processes that require prior eyeblink conditioning, were affected in these five mouse lines (*Figure 3—figure supplement 1*). After training, 4 days of extinction led to the near-disappearance of CRs in all autism model groups (CR percentage, day 12 acquisition vs day 4 extinction; paired t-tests, p < 0.05 for all comparisons) except for *L7/Pcp2$^{Cre}$::Tsc1$^{flox/flox}$*, which did not acquire CRs in the first place. The time courses of extinction were

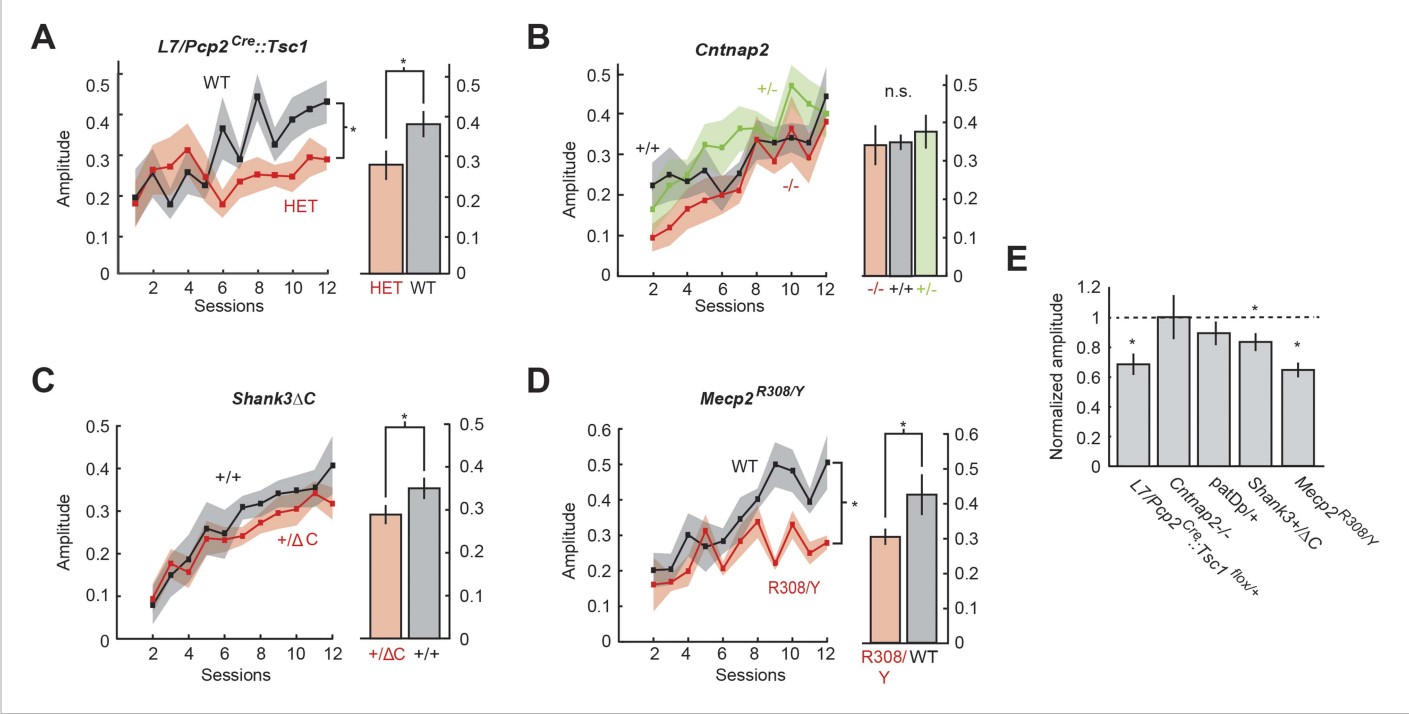

**Figure 4**. Amplitude defects are present in three mouse models. (**A**) Time course of response probability with acquisition training in L7-Tsc1 model mice. Black: WT. Red: *L7/Pcp2^Cre^::Tsc1^flox/+^*. (**B**) Time course of response probability with acquisition training in *Cntnap2* model mice. Black: *Cntnap2+/+*. Red: *Cntnap2−/−*. Green: *Cntnap2+/−*. (**C**) Time course of response probability with acquisition training in *Shank3*ΔC. Black: *Shank3+/+*. Red: *Shank3+/*ΔC. (**D**) Time course of response probability with acquisition training in *Mecp2^R308^*. Black: WT. Red: *Mecp2^R308/Y^*. In panels (**A**) through (**D**), bar plots indicate response probability averaged over the last four training sessions. (**E**) Probability deficits across all groups. Dashed line: normalized wild-type littermate level. In all panels, shading and error bars indicate SEM, and * indicates p < 0.05. n ≥ 10 mice for each group.

not statistically distinguishable between any autism model group and its corresponding wild-type littermates (p > 0.05 for all main genotype effects), indicating that perturbation of cerebellar cortex-dependent and other mechanisms that are necessary for initial eyeblink conditioning (*Robleto et al., 2004*) did not strongly affect overall extinction in the mouse models. In addition, the mouse models that initially acquired CRs also successfully reacquired CRs after 3 days of retraining (*Figure 3—figure supplement 1B*; paired t-tests of day 4 extinction vs day 3 reacquisition, p < 0.05 for all comparisons), with no appreciable difference in CR percentage between groups (p > 0.05 for all instances). The accelerated nature of this reacquisition, a process known as savings, is thought to depend in part on plasticity in the DCN (*Medina et al., 2001*; *Ohyama et al., 2006*). In short, learning deficits in the mouse models tested were specific to acquisition and were not observed in extinction or reacquisition.

## Defects of CR timing

The cerebellum is thought to be critical for task timing, and both patients with cerebellar lesions and autism patients show disrupted timing in cerebellum-dependent tasks, including eyeblink conditioning. We therefore examined the timing of the CRs during unpaired CS trials, for which the entire response time course could be analyzed (*Figure 5*). Two groups of mice showed differences in timing: *Shank3+/*ΔC and *Mecp2^R308/Y^*. Learned responses produced by the *Shank3+/*ΔC mice began at the same time (onset latency: 148.7 ± 4.9 ms *Shank3+/+*, vs 144.6 ± 4.4 ms *Shank3+/*ΔC, p = 0.5), rose faster (rise time: 91.8 ± 0.5 ms *Shank3+/+* vs 79.5 ± 0.3 ms *Shank3+/*ΔC, p = 0.04, d' = 0.70), and peaked earlier (peak latency: 317.5 ± 9.0 ms *Shank3+/+* vs 287.7 ± 5.8 ms *Shank3+/*ΔC, p = 0.03, d' = 1.02) (*Figure 5A*, right) compared to wild-type littermates. In *Mecp2^R308/Y^* animals, learned responses began at the same time (onset latency: 120.9 ± 4.0 ms WT vs 117.7 ± 4.9 ms *Mecp2^R308/Y^*, p = 0.6), rose more slowly (rise time: 113.2 ± 12.4 ms WT vs 158.4 ± 15.6 ms *Mecp2^R308/Y^*, p = 0.04,

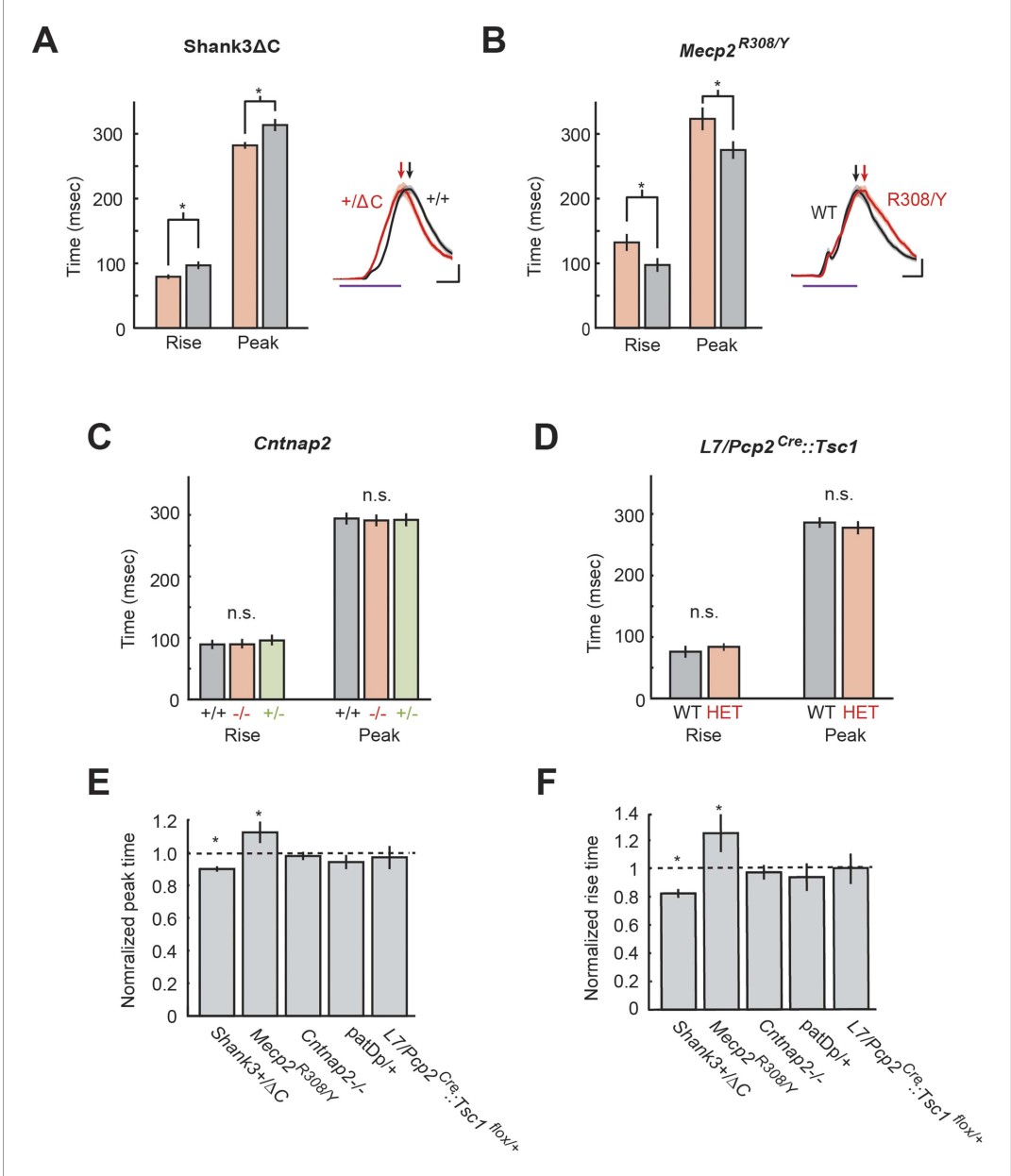

**Figure 5.** Timing defects are present in two mouse models. (**A**) Analysis of *Mecp2*[R308/Y] *Mecp2*[R308] response timing (rise time and peak latency). Inset: representative eyelid movement traces. Purple line: CS duration. Scale bars: horizontal, 100 ms; vertical, 20% of unconditioned response (UR) amplitude. Arrowheads: peak times. (**B**) Analysis of *Shank3*ΔC response timing (rise duration and peak time). Inset: representative eyelid movement traces. Purple line: CS duration. Scale bars: horizontal, 100 ms; vertical, 20% of UR amplitude. Arrowheads: peak times. (**C**) Analysis of *Cntnap2* response time (rise time and peak latency). (**D**) Analysis of L7-Tsc1 response time (rise time and peak latency) (**E**) Peak time deficits across all groups. (**F**) Rise time deficits. In plots (**E**) and (**F**), dashed lines indicate normalized wild-type littermate level. In all panels, shading and error bars indicate SEM, and * indicates p < 0.05. n ≥ 10 mice for each group.

d′ = 1.04), and peaked later (peak latency: 278.3 ± 14.9 ms WT vs 328.8 ± 16.4 ms *Mecp2*[R308/Y], p = 0.04, d′ = 1.03) compared with wild-type littermates (*Figure 5B*, right). No alterations in onset latency, peak latency, or rise time could be detected in *L7/Pcp2*[Cre]*::Tsc1*[flox/+] (*Figure 5C*), *Cntnap2−/−* mice (*Figure 5D*), or patDp/+ mice (*Piochon et al., 2014*) (p > 0.05 for all comparisons; summary of all mouse lines, *Figure 5E,F*).

## Normal sensory responsiveness

Autism has been suggested to be a general disorder of excessive sensory responsiveness, a concept known as the 'intense world' hypothesis (*Markram and Markram, 2010*). Potentially, our results in these mouse models could be accounted for by alterations in sensory responsiveness, a common feature of autism. To test this possibility, we measured responses to the US and to the pre-training CS. In US-only trials, we found no differences in unconditioned response (UR) latency measured from US onset (p ≥ 0.2 for unpaired comparisons for each cohort) or UR rise time (p ≥ 0.4 for unpaired two-sample comparisons for each cohort) (*Table 1*, 'Unconditioned response') and no differences in the correlation between UR velocity and UR magnitude (analysis of covariance group × peak interaction, p ≥ 0.2 for all cohorts). We detected no differences among wild-type cohorts for UR latency (one-way ANOVA, p = 0.5, $F_{(4,64)} = 0.92$) or velocity (one-way ANOVA, p = 0.4, $F_{(4,64)} = 1.08$).

As a second measure of sensory processing, on the first training day we observed robust eyelid opening in response to the light CS within 100 ms of CS onset (*Table 1*, 'Eyelid opening'). Eyelid opening only occurred when animals had not yet begun to produce CRs, indicating that these responses were non-associative in nature. Eyelid opening occurred on a similar fraction of trials in all groups (p > 0.1 for unpaired comparisons between each autism model and wild-type littermates). Wild-type groups also did not differ detectably (one-way ANOVA, p = 0.9, $F_{(4,64)} = 0.22$). In summary, sensory sensitivity was unaltered in any of the mouse models, and thus, deficits in delay eyeblink conditioning were not accompanied by upstream alterations in sensory sensitivity or downstream deficits in blink capability.

## Absence of gross motor deficits

Motor impairments are common in autism patients (*Fournier et al., 2010*), and cerebellar injury leads to both acute and long-lasting motor deficits. However, past investigations of our mouse models show mild or no motor impairments except for gait alterations in patDp/+ mice (*Piochon et al., 2014*). To extend these measurements, in three mouse models we analyzed gait, a motor function that can proceed without learning. We measured forepaw stance, forepaw stride, hindpaw stance, and hindpaw stride. We observed no differences between mutant and wild-type mice in *Cntnap2−/−* mice, *L7/Pcp2^{Cre}::Tsc1^{flox/+}* mice, and *Shank3+/ΔC* mice (two-sample t-test, p > 0.05 for all comparisons; *Table 1*, 'Gait analysis'). The *L7/Pcp2^{Cre}::Tsc1^{flox/+}* result is consistent with previous reports (*Tsai et al., 2012*). Taken together with past research, our findings indicate that gross motor function in adult ASD mouse models is not a necessary consequence of disruption in cerebellum-dependent learning.

## Normal learning of a water Y-maze

Mouse models of autism have been shown to be impaired in fear conditioning and hippocampus-dependent reversal (*Crawley, 2008*; *Silverman et al., 2010*). To test a second, non-cerebellar form of learning, we subjected three of our models to initial acquisition of a water Y-maze. After four training sessions, we did not observe any statistically detectable difference in the ability to find the platform in *Cntnap2−/−* mice, *L7/Pcp2^{Cre}::Tsc1^{flox/+}* mice, or *Shank3+/ΔC* mice (two-sample t-test, p > 0.05 for all comparisons; *Table 1*, 'Swimming Y-maze acquisition'). The *L7/Pcp2^{Cre}::Tsc1^{flox/+}* finding is consistent with previous reports of normal T-maze acquisition (*Tsai et al., 2012*). Therefore, the eyeblink-conditioning deficits we have observed do not reflect a broad impairment in learning mechanisms.

## Cerebellar gross anatomy and cellular morphology

Since eyeblink conditioning depends on the cerebellum, we searched for gross anatomical and cell morphological defects in the cerebella of our mouse models. Using histological methods, in *Shank3+/ΔC*, *Cntnap2−/−*, *Mecp2^{R308/Y}*, and patDp/+ mice, we found no differences between mutant mice and wild-type littermates in PC density, anterior or posterior granule layer thickness, and anterior or posterior molecular layer thickness (p > 0.1, all comparisons). In *L7/Pcp2^{Cre}::Tsc1^{flox/+}* mice, for which alterations in PC density have been previously reported (*Tsai et al., 2012*), we found no difference for anterior or posterior granule layer thickness and molecular layer thickness for *L7/Pcp2^{Cre}::Tsc1^{flox/+}* (p > 0.1, all comparisons). In summary, with the exception of *L7/Pcp2^{Cre}::Tsc1^{flox/+}* mice, these mouse lines do not show gross alterations in granule or PC density.

**Table 1**. Normal sensory responsiveness, gross motor function, and non-cerebellar learning and memory in five autism mouse models

| | L7/Pcp2^Cre^::Tsc1^flox/+^ | Cntnap2−/− | patDp/+ | Shank3+/ΔC | Mecp2^R308/Y^ |
|---|---|---|---|---|---|
| **Unconditioned response** | | | | | |
| N | 18, 16 | 12, 13 | 10, 11 | 17, 21 | 11, 12 |
| UR latency (ms) | 31.0 ± 8.6 | 32.6 ± 3.1 | 45.9 ± 8.4 | 34.5 ± 5.9 | 42.0 ± 8.7 |
| | 29.9 ± 4.3 | 27.4 ± 5.0 | 30.7 ± 7.3 | 39.5 ± 8.2 | 43.1 ± 9.3 |
| UR rise time (ms) | 64.9 ± 4.7 | 65.7 ± 5.5 | 67.3 ± 4.3 | 62.8 ± 3.6 | 60.1 ± 5.5 |
| | 57.5 ± 3.8 | 64.8 ± 5.5 | 72.8 ± 6.3 | 62.6 ± 3.8 | 64.8 ± 6.5 |
| **Eyelid opening** | | | | | |
| N | 18, 16 | 12, 13 | 10, 11 | 17, 21 | 11, 12 |
| Amplitude (% UR amp) | 13.9% ± 3.9% | 6.4% ± 1.2% | 13.4% ± 4.8% | 11.8% ± 3.1% | 13.4% ± 5.5% |
| | 15.6% ± 5.0% | 11.1% ± 3.0% | 11.8% ± 7.8% | 9.3% ± 2.6% | 13.7% ± 5.8% |
| **Gait analysis** | | | | | |
| N | 6, 7 | 10, 10 | – | 5, 4 | – |
| Fore stride (cm) | 4.61 ± 0.21 | 5.01 ± 0.21 | – | 4.82 ± 0.31 | – |
| | 4.35 ± 0.14 | 5.15 ± 0.46 | – | 4.92 ± 0.28 | – |
| Fore stance (cm) | 1.42 ± 0.06 | 1.39 ± 0.14 | – | 1.84 ± 0.12 | – |
| | 1.56 ± 0.06 | 1.41 ± 0.07 | – | 1.64 ± 0.12 | – |
| Hind stride (cm) | 4.85 ± 0.27 | 5.22 ± 0.34 | – | 4.98 ± 0.27 | – |
| | 4.84 ± 0.15 | 5.09 ± 0.42 | – | 5.07 ± 0.29 | – |
| Hind stance (cm) | 2.62 ± 0.17 | 2.20 ± 0.16 | – | 2.37 ± 0.12 | – |
| | 2.69 ± 0.16 | 2.00 ± 0.17 | – | 2.27 ± 0.12 | – |
| **Swimming Y-maze acquisition** | | | | | |
| N | 6, 7 | 10, 10 | – | 5, 4 | – |
| Acq. 1 (% correct trials) | 65.7% ± 12.9% | 81.5% ± 6.3% | – | 65.0% ± 8.6% | – |
| | 76.9% ± 7.9% | 71.1% ± 11.6% | – | 52.0 ± 10.0% | – |
| Acq. 2 (% correct trials) | 90.0% ± 6.8% | 89.0% ± 7.4% | – | 61.0% ± 17.2% | – |
| | 75.6% ± 7.0% | 91.1% ± 4.8% | – | 70.0% ± 17.3% | – |
| Acq. 3 (% correct trials) | 90.0% ± 6.8% | 96.0% ± 2.7% | – | 90.0% ± 10.0% | – |
| | 80.8% ± 8.2% | 95.6% ± 3.0% | – | 95.0% ± 5.0% | – |
| Acq. 4 (% correct trials) | 80.0% ± 20.0% | 98.0% ± 2.0% | – | 100% ± 0% | – |
| | 90.0% ± 5.7% | 100% ± 0% | – | 94.3% ± 3.7% | – |
| Test (% correct trials) | 91.3% ± 4.2% | 94.8% ± 3.1% | – | 87.2% ± 7.9% | – |
| | 93.4% ± 3.3% | 99.0% ± 1.0% | – | 97.2% ± 2.8% | – |

*Unconditioned response* was measured in terms of latency and rise time. *Eyelid opening* in response to initial CS trials was scaled to the size of the unconditioned response. *Gait* was measured as stride and stance (cm) for both forepaws and hindpaws. *Swimming Y-maze acquisition* was measured in terms of percentage of correct trials over valid trials for four acquisition periods and a test period. For all cells, top value (roman text) indicates the mutant mouse, while bottom value (italic text) indicates the control or wild-type littermates. All values mean ± SEM. All paired statistical comparisons yielded p-values greater than 0.05.

UR, unconditioned response.

PC arbors are shaped by the cumulative effects of granule cell (GrC) input (*Joo et al., 2014*), and therefore, would be potentially altered in their form. We used Sholl analysis to examine the morphology of PC dendritic arbors in *Shank3+/ΔC*, *Cntnap2−/−*, *Mecp2*^R308/Y^, patDp/+, and *L7/Pcp2*^Cre^::*Tsc1*^flox/+^ mice. Only *Shank3+/ΔC* mice differed from wild type, showing higher complexity of distal dendrites (two-way repeated measures ANOVA, main genotype effect, F(1,39) = 3.50, p = 0.07), with a significant distance × genotype interaction (F(16,624) = 2.77, p = 0.0002; *Figure 6A*).

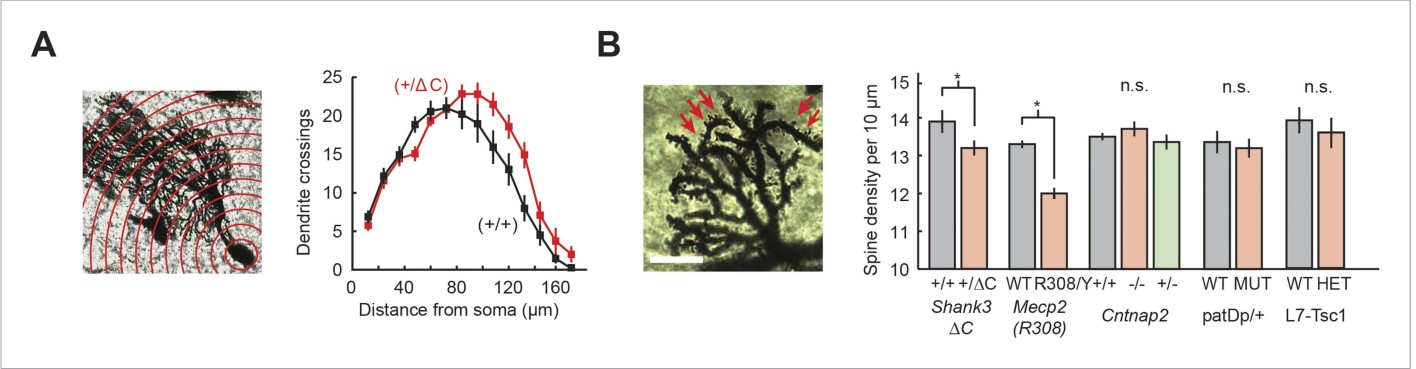

**Figure 6**. Purkinje cell dendritic arbors show structural defects in *Shank3*+/ΔC and *Mecp2*[R308/Y] mice. (**A**) Purkinje cell (PC) dendrite arborization defect is present in *Shank3*+/ΔC. Left: Sholl analysis example for *Shank3*+/ΔC. Right: groupwise Sholl analysis for *Shank3*+/ΔC. Sholl analysis for other four mouse models did not show similar arborization defects, as shown in *Figure 6—figure supplement 1*. (**B**) Spine density defects are present in Shank3+/ΔC and *Mecp2*[R308/Y]. Left: example image of *Shank3*+/+ dendritic arbor. Right: spine density for *Shank3*+/ΔC and *Mecp2*[R308/Y] groups. In all panels, shading and error bars indicate SEM, n.s. indicates p > 0.05, and * indicates p < 0.05. n ≥ 12 cells for each group.

The following figure supplement is available for figure 6:

**Figure supplement 1**. Lack of difference in PC arborization in four ASD mouse models.

Further analysis of *Shank3*+/ΔC mice revealed that compared with wild type, the center of mass of the Sholl distribution was farther from the soma (p = 0.03) and had a greater total number of crossings at distances farther than 96 µm from the soma (p = 0.01).

Closer examination of PC arbors (*Figure 6B*) revealed a decrease in the number of visible spines per 10 µm on distal dendrites of *Shank3*+/ΔC mice (p = 0.04) and *Mecp2*[R308/Y] mice (p < 0.0001). The remaining three models showed no differences in either PC arbors or spine density (*Figure 6B*, *Figure 6—figure supplement 1*; p > 0.3 for main group effect and p > 0.5 for space × genotype interactions for all comparisons; p > 0.4 for all pairwise comparisons of spine density). In summary, differences in dendritic morphology were found specifically in *Shank3*+/ΔC and *Mecp2*[R308/Y] mice, consistent with alterations in GrC input and/or PC dendritic growth mechanisms.

## Discussion

Our principal finding is that five mouse models of ASD show deficits in delay eyeblink conditioning, a learning task that requires the cerebellum (*Figure 7A*). The five models tested showed three major categories of deficit (*Figure 7B*): in the process of acquiring the CR, in the amplitude of the CR, and in the timing of the CR. Taken together, these findings paint a behaviorally based picture of how diverse ASD-related genetic conditions affect a single learning process. Together with mouse studies of neuroligin-3 (*Baudouin et al., 2012*) and Fragile X mental retardation 1 (*Koekkoek et al., 2005*) and a valproate rat model of autism (*Stanton et al., 2007*; *Murawski et al., 2009*), our work brings to eight the number of autism rodent models with alterations in cerebellum-dependent function.

Delay eyeblink conditioning is a more precise assay of cerebellar function than two phenotypes that are commonly assumed to measure cerebellar function, rotarod and gait. Rotarod and gait can reveal malfunction in a wide range of brain structures, including cerebellum (*Thach and Bastian, 2004*), striatum (*Rothwell et al., 2014*), and basal ganglia (*Takakusaki et al., 2008*). In contrast, delay eyeblink conditioning (as well as another form of learning, vestibulo-ocular reflex gain modulation) has well-mapped relationships to cerebellum and brainstem circuitry (*Raymond et al., 1996*; *Boele et al., 2010*). Our findings suggest specific cerebellar circuit elements that can be investigated further, either in non-human animals or in autistic patients.

### Cerebellar circuitry underlying eyeblink-conditioning parameters

Our conditioning experiments quantified dysfunction in two tasks for which the cerebellum is well-suited: associative learning between multiple senses and the detection of fine timing differences.

**Figure 7.** Cerebellar learning and performance deficits co-vary with circuit-specific gene expression patterns. (**A**) The first four data columns show perturbations in learning (green shading) and performance (yellow shading). The last three columns show combined gene expression (*Figure 1*) and morphological (*Figure 5*) perturbations for the olivocerebellar (red shading) and granule cell layer (blue shading) pathways, along with extracerebellar (dark gray) pathways. Note that *Cntnap2+/−*, which has been reported to be not behaviorally different from *Cntnap2+/+* (*Peñagarikano et al., 2011*), is shown for reference. *Table 2* is an expanded tables of the phenotypes described here. (**B**) Response amplitude and probability in transgenic mice (open circles) normalized to wild-type littermate ('WT') means for all models. Dark gray shading indicates mutants for which there were also timing defects. Error bars indicate SEM. (**C**) The canonical cerebellar circuit. Input along the CS (turquoise) pathway via mossy fibers (mf) from the pontine nuclei enters the cerebellar cortex through granule cells (GrC), which receive feedforward and feedback inhibition from Golgi cells (GoC) in the granule cell layer. GrCs send parallel fiber (pf) projections to PC dendritic arbors. PCs also receive teaching signals along the US (gray) pathway via climbing fibers (cfs) from the inferior olive. The output of clustered PCs (gray) converges onto neurons in the deep cerebellar nuclei (DCN), which drive downstream neurons in the output pathway.

The following figure supplement is available for figure 7:

**Figure supplement 1.** Expression patterns of ASD model genes in cerebellum.

Two pathways—the olivocerebellar loop (*Figure 7C*, red pathway) and the GrC layer input pathway (*Figure 7C*, blue pathway)—play key roles in the acquisition of learned eyeblink responses in mammals (*McCormick and Thompson, 1984*; *Yeo and Hesslow, 1998*; *Garcia et al., 1999*; *Attwell et al., 2001*; *Longley and Yeo, 2014*), including mice (*Koekkoek et al., 2003*). Information about the aversive US is conveyed through the olivocerebellar loop, consisting of PCs in the cerebellar cortex, the inferior olive, and the DCN (*Figure 7C*, red pathway). This information instructs plasticity in the mossy fiber (mf)—GrC—PC pathway, which conveys incoming CS information. The GrC layer pathway undergoes multiple forms of plasticity, including parallel fiber (PF)-PC long-term depression (*Hansel et al., 2001*; *Carey and Lisberger, 2002*; *Gao et al., 2012*), and after training. PC output helps to drive a well-timed and well-formed CR (*Choi and Moore, 2003*) and drive late-stage plasticity in the DCN (*Zheng and Raman, 2010*). Thus, defects in the reliable learning and production of CRs might be interpreted as disruption of the olivocerebellar 'instruction' system (*Garcia et al., 1999*) or the granule cell layer 'representation' system (*Arenz et al., 2009*).

Activity in the GrC network, which receives direct mf input, is thought to represent key temporal components to drive a well-timed response (*Medina and Mauk, 2000*; *D'Angelo and De Zeeuw, 2009*). Because PC sodium-based simple-spike output acts as an approximately linear readout of synaptic drive (*Walter and Khodakhah, 2006*), the time course of CR production might be expected

to be constructed from summed patterns of activity in specific combinations of GrCs and inhibitory neurons. Therefore, defects in response timing and amplitude might be interpreted as disruption of synaptic transmission and/or plasticity in the MF pathway.

## Putative substrates for learning defects: climbing fiber signals and PC excitability

Four mouse models showed decreases in the CR probability: L7-Tsc1 (*L7/Pcp2*$^{Cre}$*::Tsc1*$^{flox/+}$ and *L7/Pcp2*$^{Cre}$*::Tsc1*$^{flox/flox}$), patDp/+, *Cntnap2*–/–, and *Shank3*+/ΔC. Upon investigating patterns of gene expression, we found that the disrupted genes in three models (L7-Tsc1, patDp/+, and *Cntnap2*–/–) are expressed in PCs, inferior olive, and/or DCN (*Figure 7A*, light green and red [regular case], respectively; *Figure 7C*, red).

In *L7/Pcp2*$^{Cre}$*::Tsc1*$^{flox/+}$ mice, which are PC-specific, early-life loss of Tsc1 leads to increased spine density and decreased excitability in PCs (*Tsai et al., 2012*). This decreased excitability can affect learning by interfering with climbing fiber (cf)-based instruction, either by reducing PC dendritic excitability or by making the cerebellar cortex less effective at influencing the DCN, resulting in inhibited IO responsiveness to the US (*Schonewille et al., 2010*). Reduced PC firing would also be expected to reduce response amplitudes, which we have observed. Similarly, patDp/+ mice show cf structural plasticity during development and deregulated PF-PC LTD in adults (*Piochon et al., 2014*), echoing findings in other models (*Koekkoek et al., 2005*; *Baudouin et al., 2012*). It should be noted that other forms of cerebellar plasticity can contribute to learning in the absence of PF-PC LTD (*Schonewille et al., 2011*). Taken together, the evidence suggests that cerebellar learning defects in autism mouse models may be strongly shaped by reduced function in the olivocerebellar circuit and associated synaptic plasticity mechanisms.

The fourth model that showed a probability defect was *Shank3*+/ΔC. *Shank3* is expressed specifically at postsynaptic densities in the granule cell layer in the mouse cerebellum (*Tu et al., 1999*; *Böckers et al., 2004*, *2005*). We observed increased elaboration of the distal dendrites along with decreased spine density (*Figure 6*; *Figure 7A*, light green and red cells [bold case]). Neurotrophin-3 (NT-3) from GrCs is required for PC dendritic morphogenesis (*Joo et al., 2014*), suggesting the possibility that the *Shank3*+/ΔC mutation may disrupt PC dendritic function.

## Putative substrates for performance defects: the granule cell pathway

We observed both amplitude and timing defects in two mouse models (*Figure 7B*, gray circles), *Shank3*+/ΔC and *Mecp2*$^{R308/Y}$. These genes are expressed in GrCs (*Figure 7A*, yellow and turquoise cells, respectively), and *Mecp2* is also expressed in Golgi cells (GoCs). *Shank3* encodes a scaffolding protein that may influence MF-GrC and GrC-PC synaptic function by reducing glutamatergic transmission and plasticity (e.g., *Peça et al., 2011*; *Yang et al., 2012*; *Kouser et al., 2013*), thus, impairing cerebellar learning (*Giza et al., 2010*; *Andreescu et al., 2011*). Likewise, *Mecp2* expression is dramatically upregulated in GrCs after P21, a time when MF-GrC and PF-PC synapses are formed and still maturing (*Altman, 1972*), suggesting that Mecp2 plays a role in MF-GrC synapse function (*Mullaney et al., 2004*) and glutamatergic synaptic transmission and plasticity (*Moretti et al., 2006*). It is notable that despite the fact that *Mecp2* is also expressed in PCs (*Mullaney et al., 2004*), *Mecp2*$^{R308/Y}$ mice showed no defect in probability of learning. We chose these mice for their relatively weak motor dysfunction so that we could characterize eyeblink-conditioning deficits in detail. Other *Mecp2* mutants might show more of a probability phenotype.

## Extracerebellar sites

In addition to specific cerebellar substrates, delay eyeblink conditioning also depends on processing outside the cerebellum (*Boele et al., 2010*; *Figure 7A*, dark gray cells; *Figure 7C*, dark gray arrows). Several genes in our mouse models (though not *Shank3*) are likely to be expressed in trigeminal nucleus, which transmits sensory information to the pons and mf pathway, as well as the red nucleus and facial nucleus, which ultimately drive the production of the eyeblink (*Figure 7A*, dark gray cells; *Figure 7C*, dark gray arrows; *Figure 7—figure supplement 1*). The acquisition of delay eyeblink conditioning may also be modulated by the amygdala and hippocampus (*Lee and Kim et al., 2004*; *Boele et al., 2010*; *Sakamoto and Endo, 2010*; *Taub and Mintz, 2010*), but we did not detect two

known consequences of such modulation, learning during the first training session and short-latency alpha responses to the CS.

## Comparison with eyeblink-conditioning phenotypes in autistic persons

Past investigations of autism (*Sears et al., 1994*; *Oristaglio et al., 2013*) and Fragile X syndrome (*Koekkoek et al., 2005*; *Tobia and Woodruff-Pak, 2009*) have reported the percentage of CS-responses that exceed a fixed threshold ('% CRs'), as well as CR size averaged across all trials. However, these measures conflate changes in the probability of learning with changes in the amplitude of learned responses. For example, a study that examined de novo (i.e., no previous conditioning) delay eyeblink conditioning (*Sears et al., 1994*) found that in high-functioning (average IQ > 100) autistics, the %CR fraction rose more rapidly than in controls, reaching close to a half-maximum after only two blocks of trials. In the direction of loss-of-function, impairments in delay eyeblink conditioning have been observed in Fragile X patients (*Koekkoek et al., 2005*; *Tobia and Woodruff-Pak, 2009*); in this case, PC-specific knockout of the Fragile X protein Fmr1 in mice was sufficient to cause eyeblink-conditioning defects, suggesting that learning was specifically perturbed. For comparison with the work reported here, future human eyeblink-conditioning studies would have to distinguish changes in learning from changes in response amplitude.

A second promising domain for investigations of ASD patients is eyeblink response kinetics. Variations in response kinetics may depend on the specific genetic background. In idiopathic autism (*Sears et al., 1994*), CRs came approximately 50 ms earlier, as measured using both the time to CR onset and the time to CR peak. Similarly, after two sessions of trace conditioning (*Oristaglio et al., 2013*), delay conditioning initially leads to a decrease in response onset and latency of approximately 50 ms, followed by a convergence toward normal performance as training continues. In contrast, Fragile X patients show no differences in timing in early training sessions (*Koekkoek et al., 2005*; *Tobia and Woodruff-Pak, 2009*), but after average CR amplitude reaches a plateau, the peak latency to CR decreases by approximately 30 ms (*Tobia and Woodruff-Pak, 2009*). Changes of 30–50 ms are comparable in size to the effects we have observed in mice with granule cell pathway perturbation. In addition, in a valproate-based rat model of autism (*Arndt et al., 2005*), prematurely timed eyeblink responses were found for long interstimulus intervals (*Murawski et al., 2009*). In summary, past findings suggest that perturbation of cerebellar granule cell layer activation may be common in both idiopathic and syndromic autism. The general observation of shortened latency is consistent with our findings in *Shank3+/ΔC* mice, suggesting this line as a model for the timing deficits observed in autistic persons.

Finally, although past measurements have been done in older children post-diagnosis, eyeblink conditioning can be assayed in subjects as young as 5 months of age (*Claflin et al., 2002*). The possibility of early testing suggests that delay eyeblink conditioning could be a biomarker (*Reeb-Sutherland and Fox, 2015*) for identifying pre-diagnosis perturbations in cerebellum-dependent learning.

## The cerebellum in cognition and autism

Eyeblink-conditioning defects appear more often in mouse autism models than other non-autism-like phenotypes (*Table 2*). This specific dissociation (i.e., the absence of correlation with non-cerebellar phenotypes) suggests that cerebellar plasticity and autism's cognitive deficits might be related in some specific manner. The cerebellum arises repeatedly in the study of autism (*Wang et al., 2014*). In an analysis of gene–phenotype associations (*Meehan et al., 2011*), autism-related genes were found to be associated with a cluster of phenotypes that included social defects, abnormal motor behavior, and cerebellar foliation. A number of ASD genes are co-expressed in the cerebellum (*Menashe et al., 2013*), and ASD patients show differences in many cerebellar cell types (*Bauman and Kemper, 1985*; *Fatemi et al., 2002*; *Whitney et al., 2008*; *Wegiel, et al., 2010*) as well as gross cerebellar structure, starting at an early age (*Hashimoto et al., 1995*; *Abell et al., 1999*; *Stanfield et al., 2008*; *Courchesne et al., 2011*). Therefore, ASD genes are highly likely to shape cerebellar circuit function. Effects on cerebellar function could even have downstream consequences for function of distal brain regions of known cognitive significance to which the cerebellum supplies information (*Wang et al., 2014*).

However, our results must also be reconciled with a recent study that started not from ASD genes, but from specific perturbations to cerebellar function (*Galliano et al., 2013*). That work revealed little

**Table 2.** Complete table of previously reported autism-like and motor defects in mouse models combined with data from the present study

| Mouse model | Autism-like behaviors | | | | Delay eyeblink conditioning | | | Movement/strength | | Other tasks | | | |
|---|---|---|---|---|---|---|---|---|---|---|---|---|---|
| | Social calls | Ultrasonic calls | Grooming time | Maze flexibility | Eyeblink learning* | Eyeblink amplitude* | Eyeblink timing* | Rotarod | Gait | Maze acquisition | Startle and prepulse inhibition | Anxiety | Learned fear |
| Shank3 [a] | ↓ | ↑ | ↑ | nd | → | → | → | ↔ | ↔* | ↔* | → | nd | ↔ |
| Cntnap2 [b] | ↓ | ↓ | ↑ | → | → | ↔ | ↔ | ← | ↔* | ↔* | ↔ | ↔ | nd |
| Mecp2 [c, d, e] | ↓ | ↓ | nd | nd | ↔ | → | ← | ↔ | nd | → | nd | ← | → |
| L7-Tsc1–mutant [f] | ↓ | ↓ | ↑ | → | → | nd | nd | → | → | ↔ | nd | nd | nd |
| L7-Tsc1–het [f] | ↓ | ↓ | ↑ | ↓ (?) | → | → | ↔ | ↔ | ↔ | ↔ | nd | nd | nd |
| patDp/+ [g, h, i] | ↓ | ↓ | nd | → | → | ↔ | ↔ | ← | → | ↔ | ↔ | ← | ← |

*Social*, downward arrows indicate reduced performance on three-chamber preference test of mouse vs object, interactions with novel mouse, or play behavior. Ultrasonic vocalizations (USV) are used as an assay of communicative behavior. *Ultrasonic*, downward arrows indicate longer latency or fewer calls (adult), or more distress calls or longer latency to first call by pups. Repetitive or perseverative behaviors are assayed by grooming and flexibility on maze tasks. *Eyeblink learning*, downward arrows indicate a decrease in response probability. *Eyeblink amplitude*, downward arrows indicate a decrease in response amplitude. *Eyeblink timing*, downward arrows indicate earlier shifts in peak latency and decrease in rise time, while upward arrows indicate later shifts in peak latency and increase in rise time. *Maze flexibility*, downward arrows indicate impairment on T-maze alternation or reversal or flexibility on a Morris water or Barnes maze. Gross motor functions are assayed by rotarod and gait tasks. *Rotarod*, table entries indicate differences in the time to fall from an accelerating rotarod. *Gait*, table entries indicate differences in stance or stride parameters. *Maze acquisition*, downward arrow indicates impairment of acquisition on Morris water maze, Barnes maze, walking T-maze, or swimming T-maze. *Anxiety*, up arrows indicate increased freezing and closed-arm preference in elevated plus maze, increased light–dark preference, or decreased open-field behavior. Unless otherwise specified, the downward arrow indicates a significant decrease relative to wild-type, the upward arrow indicates a significant increase relative to wild-type, the horizontal arrow indicates no significant difference relative to wild-type, and 'nd' indicates unknown. The '*' in row 5 indicates a difference lacking statistical significance. References: [a] *Kouser et al., 2011*; [b] *Peñagarikano et al., 2011*; [c] *Shahbazian et al., 2002a*; [d] *Moretti et al., 2006*; [e] *De Filippis et al., 2010*; [f] *Tsai et al., 2012*; [g] *Nakatani et al., 2009*; [h] *Tamada et al., 2010*; [i] *Piochon et al., 2014*.

effect on a variety of standard non-motor tasks, including social, navigational, and memory tasks. Those tasks differ from current tests of autism model face validity. For example, the social assay involved consecutive presentation of mouse/object stimuli, as opposed to the simultaneous choice that occurs in the three-chamber test (*Yang et al., 2011*). Likewise, no test was given for perseveration such as maze reversal or grooming duration (*Tsai et al., 2012*). We suggest that rigorous evaluation of cerebellar involvement in non-motor function will require tasks of greater difficulty and complexity than past practice.

### Subsecond sensory integration and the etiology of autism

We have shown that mouse autism models have difficulty in a cerebellum-dependent form of associating sensory stimuli that are spaced closely in time. The integration of closely timed events across sensory modalities could be critical for statistical learning. Statistical learning can encompass the association of an auditory or visual stimulus to predict some other event, a capacity that is likely to be at the core of the acquisition of language (*Ferguson and Lew-Williams, 2014*) and other cognitive capacities (*Dinstein et al., 2012*). Such learning is commonly assumed to require neocortical plasticity via Hebbian uninstructed learning. In addition, statistical learning from unexpected events is also efficiently supported by instructed plasticity (*Courville et al., 2006*), a phenomenon for which cerebellar circuit architecture is well-suited (*Marr, 1969*). Since the neocortex and cerebellum communicate with one another bidirectionally, these two brain systems might play complementary roles in learning from experience. Projections to forebrain are present in early postnatal life (*Diamond, 2000*), and early childhood disruption of the cerebellum affects the development of social cognition and language (*Riva and Giorgi, 2000*; *Steinlin, 2008*; *Bolduc et al., 2012*). In this context, eyeblink conditioning is an example of learning from the close timing of two events of different sensory modality, and defects in it may reflect broader difficulties in subsecond temporal sensory association. If such difficulties are present in early stages of autism, the cerebellum may be a potential target for early-life therapeutic intervention.

## Materials and methods

### Animals

*Cntnap2* mice were bred at Princeton University on a heterozygote–heterozygote strategy using breeding pairs obtained from the Geschwind laboratory at the University of California, Los Angeles (*Peñagarikano et al., 2011*). These animals were originally generated by the Peles laboratory (Weizmann Institute of Science, Israel) through the replacement of the first exon of *Caspr2* (*Cntnap2*) using gene-targeting techniques in mice with the imprinting control region (ICR) background (*Poliak et al., 2003*). The mice were then outbred on the C57BL/6J background for at least 10 generations and characterized behaviorally (*Peñagarikano et al., 2011*). For behavioral experiments, 39 animals from 17 litters were used.

*Shank3*+/ΔC mice were bred at Princeton University on a heterozygote–heterozygote strategy using breeding pairs acquired from the Worley laboratory at Johns Hopkins University. These mice were generated by the conditional deletion of exon 21 of *Shank3* to excise its C-terminal domain, including the Homer-binding domain (*Kouser et al., 2013*; http://jaxmice.jax.org/strain/018389.html). The mice were generated on a mixed background and backcrossed on a C57BL/6J background for at least five generations. Only heterozygotes of the C-terminal mutation were used (*Durand et al., 2007*). For behavioral experiments, 38 animals from 16 litters were used.

*Mecp2*R308/Y mice were bred at Princeton University on a heterozygote-wild-type strategy using a breeding pair acquired from Jackson Laboratories (B6.129S-Mecp2tm1Hzo/J, stock no.: 005439). Mice on the 129/SvEv background have a truncating mutation of *Mecp2* introduced through the insertion of a premature stop after codon 308 (*Shahbazian et al., 2002a*). These mice were backcrossed on the C57BL/6J background for at least 10–12 generations. Because these mice show a regressive phenotype, they were tested at 16–20 weeks, an age at which the mice begin showing cognitive symptoms and minor motor dysfunction ('early symptomatic' to symptomatic phase: *Shahbazian et al., 2002a*; *Moretti et al., 2006*; *De Filippis et al., 2010*). For behavioral experiments, 28 animals from 11 litters were used.

The Tsc1 mice were bred at Princeton University from breeding pairs on a mixed (C57BL/gJj, 129 SvJae, BALB/cJ) background acquired from the Sahin laboratory at Boston Children's Hospital,

Harvard Medical School (*Tsai et al., 2012*). These mice were originally generated by crossing *L7/Pcp2-Cre* mice with *Tsc1flox/flox* mice (*Tsai et al., 2012*). For the present study, the offspring of this cross were crossed to produce the *L7/Pcp2Cre::Tsc1flox/+* (heterozygous) and *L7/Pcp2Cre::Tsc1flox/flox* (homozygous) animals. Littermate controls were pooled from *Tsc1+/+* (pure wild-type), *Tsc1flox/+*, *L7Cre;Tsc1+/+* (L7Cre), and *Tsc1flox/flox* (flox) mice. For behavioral experiments, 34 animals from 18 litters were used.

patDp/+ (15q11-13 duplication) mice were acquired from the Hansel laboratory at the University of Chicago and the Takumi laboratory at Hiroshima University and tested as previously reported (*Nakatani et al., 2009*; *Piochon et al., 2014*). Data from the eyeblink conditioning experiments described in *Piochon et al. (2014)* are available upon request from the corresponding author.

For all experiments, we used 2- to 4-month-old males with matched littermates unless otherwise indicated. To ensure that the ages of the mice did not affect the results, we corrected our statistical tests of average CR performance, response probability, and response amplitude across sessions 9–12 and the CR timing parameters, using analysis of covariance tests with age (days) as a covariant with post hoc Tukey's tests (*Piochon et al., 2014*). This analysis produced no changes in statistical significance of the findings reported throughout this paper (Tukey's test, $p > 0.05$ in all instances).

Mice were group-housed (at $\geq$ 8 weeks of age) and maintained on a 12-hr reverse light–dark cycle with ad libitum access to food and water. All experiments were performed according to protocols approved by the Princeton University Institutional Animal Care and Use Committee.

## Eyeblink conditioning

Each mouse was head-fixed above a stationary, freely rotating foam wheel, which allowed it to locomote throughout the experiment (*Figure 2A*). In this position, the US (airpuff) could be delivered from a consistently to the eye through a blunted 27-gage needle. The eyelid deflection was detected using a Hall effect sensor (AA004-00, NVE Corporation, Eden Prairie, MN) that was mounted above the same eye (*Koekkoek et al., 2002*). Prior to placement in the experimental apparatus, each mouse was briefly anesthetized with isoflurane and a small neodymium magnet (3 mm × 1 mm × 1 mm, chrome, item N50, Supermagnetman, Birmingham, AL) was attached to the lower eyelid with cyanoacrylate glue (Krazy Glue, Westerville, OH). The sensor provided a readout of eyelid position by linearly converting a change in magnetic field due to the displacement of the magnet relative to the sensor a change in voltage. The CS (ultraviolet LED) was also delivered to the ipsilateral eye.

The animals were allowed to habituate to this apparatus for at least 195 min over 3–5 days. Following habituation, acquisition training took place over 12 training sessions (1 session/day, 6 days/week), during which the animals received 22 blocks of 10 trials each. CSs (ultraviolet light, 280 ms) were paired with an aversive US (airpuff delivered by a blunted needle to the cornea, 30–40 psi, 30 ms, co-terminating with the CS). Ultraviolet light is in the sensitive range of laboratory mice (*Jacobs et al., 2001*). Each block contained 9 paired US-CS trials and 1 unpaired CS trial, arranged pseudorandomly within the block (*Figure 2B*). Each trial was separated by an interval of at least 12 s (see below).

Following acquisition training, the mice received extinction and reacquisition training. Extinction training took place over 4 sessions (1 session/day) consisting of 22 blocks of 10 trials each. Each block contained five unpaired CS trials and five unpaired US trials, arranged pseudorandomly within the block. Reacquisition training took place over 3 sessions, and the animals received the same training sequence as in acquisition training.

## Data processing and analysis for eyeblink conditioning

Trials were triggered automatically using a custom MATLAB (Mathworks, Natick, MA) graphical user interface. Stimuli were triggered by Master-8 (AMPI, Inc., Jerusalem, Israel) via the data acquisition system (National Instruments, Austin, TX). (Scripts for data collection and analysis along with sample data are available at https://github.com/akloth0325/eyeblink-conditioning.) The Master-8 controlled the stimulus timing and sent square signals to an ultraviolet LED and a Toohey Pressure System IIe spritzer (Toohey Co., Fairfield, NJ) to generate the CS and US, respectively. The output from the Master-8 was returned to the data acquisition system. The voltage output of the Hall-effect sensor was filtered and amplified (band-pass filtered from 0.01 Hz to 4 kHz, gain adjusted to signal quality) and sent to the data acquisition system.

The beginning of an individual trial was subject to the following criteria. First, at least 12 s must have elapsed since the last trial. Time was added to the interval between any two consecutive trials

according to the stability of the eyelid position signal: if the eyelid position signal (the 'baseline' signal) strayed outside an experimenter-determined range during 1 s prior to the planned delivery of the CS, an additional 1 s was added to the intertrial interval until this criterion was met, after which the trial was initiated. The experimenter used the voltage range of UR (baseline to peak) during 3–12 unpaired US trials delivered at the beginning of the session to determine an acceptable voltage range for baseline activity prior to the beginning of each trial; typically, this range was ±10% of the average size of the UR.

The data from each trial were normalized prior to analysis. For the paired US-CS trials, the eyelid position was normalized to the range between the baseline and the peak amplitude of the UR during the trial. For the unpaired CS trials, the eyelid position was normalized to the range between the baseline and the UR peak for the most recent US-CS trial. Then, the response probability and response amplitude for a single training session were calculated. This normalization scheme yielded results that were not significantly different from those acquired by normalizing to a sessionwide average UR (paired t-tests within groups for CR performance, response probability, and response amplitude on session 12, $p > 0.05$ in all instances).

The analysis method was inspired by brain slice recording of single-synapse plasticity (*O'Connor et al., 2007*) to analyze the full range of detectable responses to a CS (*Figure 2*). The peak response size for the period between 100 ms and 280 ms after the onset of the CS was collected for every trial during each session, and a probability distribution was computed from these data. The part of the probability distribution that lay below a peak response size of 0 was considered the 'non-response distribution'. This part of the distribution plus a reflection of this distribution for a positive peak response size was subtracted from the original probability distribution. The remaining distribution was the 'response distribution'. The response probability for the given session was the area under the response distribution. The response amplitude was computed as the center of mass for the response distribution. Response timing was analyzed from the unpaired CS trials. The normalized response during the CS scored as a CR if it exceeded 0.15 between 100 ms and 400 ms after the onset of the CS and remained below 0.05 between 0 ms and 99 ms. (Again, trials for which the responses exceeded 0.05 between 0 ms and 99 ms after the onset of the CS were excluded.)

As sensory and motor tests, motor function was analyzed using unpaired US trials from the first session of training. Peak time, rise time, and onset time were calculated on smoothed individual traces as described above, within 75 ms of US onset. Photic eyelid opening was analyzed during the first session of eyeblink conditioning, during which no conditioned eyeblink was generated. Using the normalized individual eyelid deflection traces, deflections that were more than 5% below the baseline 70–250 ms after the CS onset—but not before—were counted.

## Water Y-maze acquisition

Mice underwent one session of habituation training (1 day), four sessions of acquisition training (the next day), and two sessions of testing (the following day) in a water Y-maze (custom made: 32 cm arms positioned at 120° from one another, made of semitransparent polycarbonate) filled with opaque water (non-toxic white tempera paint was added to achieve opacity). On the habituation day, mice were dropped into 10 cm of water in order to measure their swimming ability. The habituation day consisted of three 60-s trials, each trial starting from one arm of the maze. No platform was hidden beneath the surface of the water during this phase of training. During acquisition, the mice were randomly sorted into leftward-going or rightward-groups; this selection determined in which arm the platform would be hidden beneath the surface of the opaque water for each mouse. For five trials per training session, the mice were dropped into the arm closest to the experimenter and were given 40 s to find the platform. On the following day, the animals underwent two more sessions of the same protocol to test memory. The swimming trajectories of the mice were captured on video and were processed by a custom Python script (available at https://github.com/bensondaled/three-chamber) to determine whether the animal found the platform on a given trial. Excursions to the wrong arm of the maze were counted as incorrect. Results were reported the fraction of correct trials to valid trials, where valid trials included all trials on which the animal successfully to swam to either the left or the right arms of the Y-maze.

## Gait analysis

Mice videotaped during two runs along a 100-cm track over a plexiglass surface. Each run was initiated with an airpuff to the hindlimb. Runs were videotaped (iPhone 6, 40 frames/s) from below,

and light was sourced from below. After being separated using a custom MATLAB scripts, JPEG stacks were analyzed using FIJI Manual Tracker (LOCI, Madison, WI) for the centroid of each paw. Stance and stride parameters were calculated from four paw centroid trajectories (≥10 strides per run) for each animal.

## Surgery

Mice were fitted with a 1″ × ½″ × 1/32″ custom titanium headplate (*Ozden et al., 2012*; *Heiney et al., 2014*). During the surgery, each mouse was anesthetized with isoflurane (1–2% in oxygen, 1 l/min, for 15–25 min) and mounted in a stereotaxic head holder (David Kopf Instruments, Tujunga, CA). The scalp was shaved and cleaned, and an incision was made down the midline of the scalp. The skull was cleaned and the scalp margin was kept open with cyanoacrylate glue (Krazy Glue). The center of the headplate was positioned over bregma and attached to the skull with quick-drying dental cement (Metabond, Parkell, Edgewood, NY). Following the surgery, the mice received a non-steroidal anti-inflammatory drug (0.1 ml, 50 mg/ml Rimadyl [carprofen, Zoetis, Florham Park, NJ]) subcutaneously and were allowed to recover for at least 24 hr.

## Tissue processing and analysis

Tissue from separate groups of mice for each cohort was used to analyze the morphology of the cerebellum. For Nissl staining and immunohistochemistry, the mice were anesthetized with 0.15 ml ketamine-xylazine (0.12 ml 100 mg/ml ketamine and 0.80 ml mg/ml xylazine diluted 5× in saline) and transcardially perfused with 4% formalin in Delbucco's phosphate buffered saline (PBS). The brain was extracted and stored at 4°C in 4% formalin in PBS overnight. Then, the brains were split into hemispheres. The hemispheres used for Nissl staining were stored in 0.1% sodium azide in PBS at 4°C until vibratome sectioning. The hemispheres used for immunohistochemistry were prepared for cryosection. These hemispheres were stored in 10% sucrose in PBS at 4°C overnight and were blocked in a solution of 11% gelatin/10% sucrose. The block was immersed in a mixture of 30% sucrose/10% formalin in PBS for 2 hr and then stored in 10% sucrose in PBS at 4°C for up to 2 weeks.

For Golgi-Cox staining, the mice were anesthetized with 0.15 ml ketamine-xylazine (0.12 ml of 100 mg/ml ketamine and 0.80 ml mg/ml xylazine diluted 5× in saline) and decapitated immediately. The brain was removed quickly in ice-cold PBS and processed using the FD Rapid GolgiStain kit (FD Neurotechnologies, Inc., Columbia, MD), according to the kit instructions.

Brain hemispheres used for Nissl staining were blocked sectioned sagittally on a vibratome at a thickness of 70 μm. The sections were mounted on Fisherbrand SuperFrost microscope slides (Thermo Fisher Scientific, Waltham, MA) and allowed to dry at room temperature overnight. Then, they were Nissl stained with cresyl violet according to standard procedures and coverslipped with Permount (Thermo Fisher Scientific, Waltham, MA). The sections were imaged at 5× magnification and 'virtual slices' were constructed from serial images captured by the MicroBrightField software Stereo Investigator (MBF Biosciences, Williston, VT). The thicknesses of the molecular layer and the granule layer were measured on anterior and posterior portions of vermal sections of the cerebellum at 150-μm intervals using ImageJ (National Institutes of Health, Bethesda, MD).

Brains used for Golgi-Cox staining were sectioned sagittally on a vibratome at a thickness of 120 μm. The sections were mounted on slides and allowed to dry in the dark at room temperature overnight. Then, they were processed for Golgi staining according to the instructions for the FD Rapid GolgiStain kit and coverslipped with Permount. The sections were imaged at 20× and 40× and images of Golgi-stained PCs and captured by the MicroBrightField software Stereo Investigator. The cross-sectional area of the soma and the maximum height, maximum width, and the cross-sectional area of the PC dendritic arbor were measured using ImageJ. In addition, the complexity of the PC dendritic arbor was determined using Sholl analysis (*Sholl, 1956*) using ImageJ; briefly, the number of intersections of the dendritic arbor with concentric circles drawn at 12-μm intervals from the soma was counted (e.g., see *Figure 5D*). Spines on the distal dendrites were counted in an unbiased manner from these cells (e.g., see *Figure 5D*). The spines on distal dendrites of every fifth branchlet (random starting point) were counted and the dendrite length was measured.

Brain hemispheres used for immunohistochemistry were sectioned sagittally on a cryotome (−20°C) at a thickness of 30 μm and stored in PBS. Sections were immunostained with rabbit anti-calbindin (1:2000, Invitrogen, Waltham, MA) as the primary antibody and donkey anti-rabbit AlexaFluor 488

(1:300, Invitrogen, Waltham, MA). Sections were counterstained with 4',6-diamidino-2-phenylindole (DAPI, 1:100, Invitrogen, Waltham, MA). The sections were mounted on Fisherbrand SuperFrost microscope slides (Thermo Fisher Scientific, Waltham, MA) slides and coverslipped with VectaShield without DAPI (Vector Labs, Burlingame, CA). The sections were imaged at 10× magnification on an epifluorescence microscope and 'virtual slices' were constructed from serial images taken by the MicroBrightField software Stereo Investigator. PCs were counted and the length of the PC layer was measured for each sample using ImageJ.

## Statistics

All data and samples were analyzed with by an experimenter who was blinded to genotype. All pairwise statistical tests were unpaired two-sample t-tests unless otherwise noted. Time course data were analyzed using two-way ANOVAs with repeated measures; main genotype effects were reported regardless of significance, whereas main session effects (which would indicate a learning effect through time) are significant and session × genotype interactions are not significant unless otherwise indicated. When comparing a single measurement across more than two groups, one-way analyses of variance were performed with Bonferroni post hoc tests with planned comparisons. Correction for potentially confounding variables (i.e., age) was performed using analysis of covariance tests with the confounding variable as the covariant and followed by Tukey's post hoc tests. Tests were performed using GraphPad Prism 6 (GraphPad Software, Inc., La Jolla, CA) and SPSS 21 (IBM, Armonk, NY). All data are displayed as mean $\pm$ standard error of the mean (SEM) unless otherwise noted in the text or legend. Where significant differences were discovered with pairwise comparisons, effect sizes are also reported as Cohen's $d'$.

## Acknowledgements

This work was supported by R01 NS045193 (SSHW), the Nancy Lurie Marks Family Foundation (SSHW and MS), Simons Foundation Autism Research Initiative awards (221582 to SSHW and 203507 and 311232 to CH), New Jersey Commission on Brain Injury Research CBIR12FE1031 (AG), a Brain Research Foundation grant (CH), Autism Speaks (MS) a CREST grant from the Japanese Science and Technology Agency, and a KAKENHI grant from the Ministry of Education, Culture, Sports, and Technology of Japan (TT) and F31 MH098651 (ADK). The authors thank JF Medina, I Ozden, and T Schoenfeld for discussions and advice, R Jones for technical support, B Deverett for software and image analysis assistance, and LA Lynch for animal and histology support.

## Additional information

### Funding

| Funder | Grant reference | Author |
| --- | --- | --- |
| National Institute of Neurological Disorders and Stroke (NINDS) | R01 NS045193 | Samuel S-H Wang |
| National Institute of Mental Health (NIMH) | F31 MH098651 | Alexander D Kloth |
| Simons Foundation | Autism Research Initiative (221582) | Samuel S-H Wang |
| State of New Jersey Department of Health | New Jersey Commission on Brain Injury Research (CBIR12FE1031) | Andrea Giovannucci |
| Nancy Lurie Marks Family Foundation | | Mustafa Sahin, Samuel S-H Wang |
| Brain Research Foundation (BRF) | | Christian Hansel |
| Ministry of Education, Culture, Sports, Science, and Technology (MEXT) | CREST grant | Toru Takumi |

| Funder | Grant reference | Author |
|---|---|---|
| Ministry of Education, Culture, Sports, Science, and Technology (MEXT) | KAKENHI grant | Toru Takumi |
| Simons Foundation | Autism Research Initiative (203507) | Christian Hansel |
| Autism Speaks | | Mustafa Sahin |
| Simons Foundation | Autism Research Initiative (311232) | Christian Hansel |

The funders had no role in study design, data collection and interpretation, or the decision to submit the work for publication.

## Author contributions

ADK, Conception and design, Acquisition of data, Analysis and interpretation of data, Drafting or revising the article, Contributed unpublished essential data or reagents; AB, AL, AC, SGC, AG, Acquisition of data, Analysis and interpretation of data; MAB, PFW, Contributed *Shank3ΔC* mice and supplied technical information about gene expression and mouse behavior for manuscript; GG, Contributed patDp/+ mice, processed and supplied tissue for anatomical analysis, and supplied technical information about mouse line for manuscript; OP, Contributed *Cntnap2* mice and supplied technical information about mouse line for manuscript; CP, CH, Contributed patDp/+ mice, supplied tissue for anatomical analysis, supplied technical information about gene expression and mouse behavior for manuscript, and provided crucial editorial input for the drafting of the manuscript; PTT, Contributed L7-Tsc1 mice and supplied technical information about mouse behavior for manuscript, and provided crucial support during drafting of manuscript; DHG, Contributed *Cntnap2* mice and supplied technical information about gene expression and mouse behavior for manuscript; MS, Contributed L7-Tsc1 mice and supplied technical information about about gene expression and mouse behavior for manuscript; TT, Contributed patDp/+ mice and supplied technical information about gene expression and mouse behavior for manuscript; SS-HW, Conception and design, Analysis and interpretation of data, Drafting or revising the article

## Author ORCIDs

Alexander D Kloth, http://orcid.org/0000-0001-8119-1213
Giorgio Grasselli, http://orcid.org/0000-0002-2942-9615

## Ethics

Animal experimentation: All experiments were performed according to protocols (#1943-13) approved by the Princeton University Institutional Animal Care and Use Committee. All surgery was performed under isoflurane anesthesia, and every effort was made to minimize suffering.

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
