## [Decision Letter]

Thank you for sending your work entitled “Convergence of multiple mouse autism models on shared multisensory learning phenotypes” for consideration at *eLife*. Your article has been evaluated by a Senior editor, a Reviewing editor, and three reviewers, and we would like to invite you to submit a revised manuscript.

The Reviewing editor and the reviewers discussed their comments before we reached this decision, and the Reviewing editor has assembled the following comments to help you prepare a revised submission.

This manuscript describes delay eyeblink conditioning in five mouse models of autism spectrum disorder. The authors point out that the cerebellar malformations are characteristic of autism and that this behavioral task depends on the cerebellum. They therefore test whether the mutation of autism linked genes alters the acquisition and execution of conditioned eyeblink responses, and demonstrate that this is indeed the case. On the one hand, one could argue that the results are obvious – if the genes are expressed in cerebellar cells (known) and the gene products have any effects on the neurons that express them (likely) then it is inevitable that changes in eyeblink conditioning will occur. On the other hand, a characterization of the specific deficits is informative, and since there has been resistance among autism experts to acknowledge a role for the cerebellum, it is valuable to gather such survey data to make the case convincingly.

Overall, the Reviewing editor and referees agreed that this is an interesting, and potentially important study that could have a significant impact among cerebellum and autism researchers. The experiments are for the most part carefully carried out and well-presented.

However, there were serious concerns about the validity of the conclusions that the authors draw from their data, as well as their presentation of the results. Addressing these concerns will require additional experiments and quantification to strengthen the association with ASD. In particular, they should test whether the severity of social deficits is correlated with severity of impairment in eyelid conditioning – this would significantly strengthen the paper. Also, if eyelid conditioning is a more sensitive assay than other cerebellar tasks, this would provide more support for their central hypothesis. Furthermore, the authors should drop their vague claim that eyelid conditioning is a model for “multisensory learning” deficits in autism. Instead, the authors should be more direct and realistic about the content and limitations of the observations. This involves including more discussion and/or data related to timing deficits or lack thereof and how this relates specifically to literature regarding eyelid conditioning in ASD patients. The Reviewing editor and referees agreed that final acceptance or rejection of the revised manuscript will depend on the outcome of the new experiments.

1) Is there any cerebellar test which is not multisensory? If so, please indicate which one and then subject one or two mutants to such a task, so as to increase the impact of the association created.

2) Is there any non-cerebellar test which is not multisensory? If so, please indicate which one and then subject one or two mutants that express the relevant genes in the non-cerebellar regions involved to such a task, so as to increase the impact of the association created. This might create a double dissociation and/or association.

3) The deficits observed in the mice do not appear to closely resemble those observed in autistic children or adults. Sears et al. reports deficits in appropriate timing of the eyelid responses, but faster learning and larger amplitude CRs in ASD patients. Another study (Oristaglio et al.) in children with ASD also reports timing deficits. The present manuscript does not present much data on timing of CRs, save some very small changes shown in Figure 6. Instead, the amplitude and probability of the CRs is smaller in some of the mice. It is not clear how to relate these differences to those observed in humans or how they fit with the well-documented role of cerebellar cortex in controlling the timing of CRs. This is a serious concern which needs to be discussed.

4) Altered gene expression is global in 4 of the 5 mouse models used. Given this, it is unclear how the authors can be sure the deficits are actually due to changes in the cerebellum. Relatively small differences observed in amplitude and probability could have many origins outside of cerebellum. This is a major shortcoming which needs to be addressed, as one of the main motivations of the paper seems to be to approach the problem of understanding the link between cerebellum and ASD.

5) The phenotypes are mostly relatively subtle, but at the same time compelling in that all ASD mouse models show some deficit. To what extent is the eyeblink test specific for autism? Wouldn't all specific cerebellar tests show some phenotype in the ASD mouse models tested here? Please note that tests like rotarod are not cell specific.

6) To strengthen the association between cerebellar behavioral deficits and autism, shouldn't the level of the behavioral cerebellar motor deficit be correlated quantitatively to the level of more authentic autistic behavior such as grooming, vocalization, repetitive behavior etc? If there is a positive relation between these levels, the association will be stronger. So please try to provide these correlations at a quantitative level (if necessary by digging into the literature).

7) Are there mouse mutants that do show cerebellar deficits, but no signs of autism (as described above)? Please indicate and/or show which ones.

8) Please provide a more detailed description of the precise cellular distribution of the expression of all five genes involved within the entire olivocerebellar system as well as the non-cerebellar regions involved in eyeblink conditioning (e.g. trigeminal and facial nucleus and red nucleus) in the Introduction, Methods or Results section (a table would also be nice), so the reader can better relate the phenotypes later on to the anatomy. The authors already did quite a good job on this point, but it can be made more extensive, which in this case is relevant as the models concern mostly global mutants.

9) What is the point was of measuring probability in two different ways, first as 15% of the UR and next from the integration of the non-failure distribution? The two analyses did not appear to give statistically different outcomes, and it remains unclear what was gained by the second analysis (or retaining the first analysis). After the description of the method and the report that the data are in Figure 3, nothing is stated as being learned. What is the purpose of including both?

10) Results (subsection headed “General eyeblink conditioning defects”): “These findings indicate that delay eyeblink conditioning can be used to further probe defects specific to the initial acquisition of conditioning, a process that requires the cerebellum.” This concluding statement from the primary experiment of manuscript does not make sense. The first half (conditioning can be used to probe conditioning) is tautological, and the second part (cerebellar dependence) was known before the experiments were done. The sentence also doesn't follow logically from the observation that extinction was normal and acquisition was altered. Statements like this give the sense that this is an archival set of measurements that were definitely worth making and may be of value in a deeper investigative context, but that there really are no major insights gained from the work.

11) It makes sense to measure both response probability and amplitude, since these may reflect different aspects of the learning, but the presentation of the data is bizarre and makes it hard to follow the point. In the subsection headed “Performance differences associated with olivocerebellar pathway perturbations”, it is almost as though another writer took over the paper. The text reverts to explaining the circuit, restates the purpose of the experiments, re-identifies the genes, and re-calls Figure 1. All the data that actually matter in the manuscript are in Figures 4 and 5 (and the raw traces of Figure 2 are helpful as well as illustrations of data that gave rise to these summaries). The plots of Figures 4 and 5 catalogue the defects associated with different genes (useful) but are not interpreted in a way that is particularly compelling. In the last paragraph of the aforementioned subsection comes the conclusion of three of the models, “Taken together, [these] models suggest that Purkinje cell disruption is accompanied by deficits in learning.” This is not a new discovery; it is a confirmation of something long known. It seems the cart is before the horse here; it is plausible that the discovery is that these autism-linked mutations alter the cerebellar circuit sufficiently to affect learning processes known to depend on the cerebellum. That the effect can be localized to Purkinje cells, however, is not convincing.

12) Ascribing the changes in amplitude in the *MeCP2* and *Shank* mutants to the “mossy fiber pathway,” which means granule and Golgi cells, is not convincing. For *Shank*, given that the protein is expressed in the cerebellum only in granule cells (Figure 1), any changes in these cells may well contribute to deficits, but why this should affect amplitude specifically is not clearly justified. For the *MeCP2* mutants, the more widespread expression does not make the link to the mossy fiber pathway make sense. Besides, as defined, the mossy fiber pathway is everything downstream from the mossy fibers, which encompasses the full cerebellum. It should also affect Purkinje cell activity, which the authors just said would necessarily affect response probability. A key piece of information missing is what the expression of all proteins is in mossy fibers, which are heterogeneous. Besides, without knowing what the nature of the signals made by Purkinje cells is under the different conditions, these types of conclusions cannot be drawn with certainty (this does not require redoing all the experiments with electrophysiological recordings; just being realistic in the written interpretations).

[Editors' note: further revisions were requested prior to acceptance, as described below.]

Thank you for resubmitting your work entitled “Cerebellar multisensory learning defects in five mouse autism models” for further consideration at *eLife*. Your revised article has been favorably evaluated by a Senior editor, a Reviewing editor, and three reviewers. The manuscript has been improved but there are some remaining issues that need to be addressed before acceptance, as outlined below.

The referees appreciated the effort the authors have made to address their concerns and agreed that the manuscript is much improved. However, it was felt that the authors need to go further in order to tone down the rhetoric in the discussion regarding eyelid conditioning representing a model system for multisensory learning. It is also strongly recommended that the authors drop “multisensory” from the title. Detailed suggestions for revision from the three reviewers are below.

*Reviewer #1*:

Job well done. All my major concerns have been addressed.

*Reviewer #2*:

Though I appreciate the authors' efforts and believe that the data in this paper will be useful for future investigations into the intriguing links between cerebellum and ASD, I still have concerns about the overall significance of the findings as well as the authors’ interpretation. Several other papers (62; 82) have already documented deficits in eyelid conditioning in rodent models of autism and related disorders. These include a recent paper in Nature Communications from Piochon et al. (with some of the same authors as Kloth et al.) that documents eyelid conditioning deficits in one of the same mouse lines that is used in the present study. While it is true that the present manuscript shows that eyelid conditioning deficits are common to several different autism models, there doesn't seem to me any compelling pattern or mechanistic explanation for the various deficits that the authors observe in the various models. The facts that (1) the deficits are fairly subtle and vary across the different lines and (2) many of the disrupted genes are expressed outside the cerebellum and/or at multiple sites within the cerebellum make it impossible to link the phenotypes with known models/mechanisms of cerebellar learning. The authors’ attempts to make such links in the Discussion were not convincing.

I also am still unconvinced that eyeblink conditioning is a paradigm for studying multisensory deficits in autism. The authors state in the Discussion that “Learning that an auditory or visual stimulus predicts some other sensory event is likely to be at the core of how infants gain cognitive capacities. In this context, eyeblink conditioning is unusually interesting because it captures a central component of statistical learning, the close timing of two events. Future research may identify parallels between delay eyeblink conditioning and other forms of learning that rely on the cerebellum.”

This kind of “statistical learning” is what one imagines is going on in cerebral cortex. For that matter, associating two closely spaced events could occur almost anywhere in the brain via Hebbian plasticity. One thinks about the cerebellum as a structure for supervised learning or learning from error. Virtually all real-life learning is multisensory, so I really don't see the specific connection to the cerebellum. Deficits in cerebellar function may well be playing a special/critical role in autism, as highlighted in Dr. Wang's excellent recent review. The idea of disrupted internal model learning is particularly intriguing in relation to the cerebellum-autism connection. However, I am not convinced by the multisensory learning/eyelid conditioning angle pushed here.

Another concern voiced in my original review related to the unclear relationship between the eyelid conditioning deficits in mice and those reported previously in individuals with ASD. I appreciate that methodological differences complicate comparisons, but the fact that there is not a close alignment of the animal and human studies remains concerning to me.

In the first round of reviews one of the reviewers commented that “…statements like this give the sense that this is an archival set of measurements that were definitely worth making and may be of value in a deeper investigative context, but that there really are no major insights gained from the work.” This statement sums up my overall feeling about this paper.

*Reviewer #3*:

To my reading, the authors have done a very good job in revising the manuscript. It is much more coherently written, accessible, and complete. The new data lend strength to the interpretations, and the reorganization and rewriting make it much easier to understand the authors' main line of reasoning. The data present a good case for how cerebellar learning is affected in specific ways by autism linked mutations. My comments are mostly minor.

1) Introduction, first paragraph. “Sensory responsiveness and social symptoms are strongly correlated in high-functioning autism patients.” It is not clear what is meant by “social symptoms” or how they would vary along an axis.

2) Figure 1, please define and explain the red and blue (portions of the) records. I think blue is CS only in a trained animal but it is not explicitly stated.

3) In the Results, subsection “Normal extinction and reacquisition of conditioned responses”, the authors state: “…eyeblink extinction and savings, two processes ascribed to the deep cerebellar nuclei…”. This statement is inaccurate, probably because it is written as shorthand. Extinction requires the cerebellar cortex (and the whole olivocerebellar loop) as shown in Medina Nores and Mauk 2002 Nature. It is argued that savings results from plasticity in the deep cerebellar nuclei, but re-acquisition of well-timed responses (facilitated by savings) likely requires the cerebellar cortex ([76] J Neurosci). Please edit for accuracy. (Also eyeblinks are not extinguished; conditioned responses are extinguished.)

---

## [Author Response]

*[…] However, there were serious concerns about the validity of the conclusions that the authors draw from their data, as well as their presentation of the results. Addressing these concerns will require additional experiments and quantification to strengthen the association with ASD. In particular, they should test whether the severity of social deficits is correlated with severity of impairment in eyelid conditioning – this would significantly strengthen the paper. Also, if eyelid conditioning is a more sensitive assay than other cerebellar tasks, this would provide more support for their central hypothesis. Furthermore, the authors should drop their vague claim that eyelid conditioning is a model for “multisensory learning” deficits in autism. Instead, the authors should be more direct and realistic about the content and limitations of the observations. This involves including more discussion and/or data related to timing deficits or lack thereof and how this relates specifically to literature regarding eyelid conditioning in ASD patients. The Reviewing editor and referees agreed that final acceptance or rejection of the revised manuscript will depend on the outcome of the new experiments*.

We are grateful that the reviewers consider this paper a potentially significant contribution to two fields that seem to be on a collision course: ASD and the cerebellum. We have carefully taken their advice into account. We have reorganized the paper to emphasize the results before putting them into the context of cerebellar function. We have also carefully laid out how eyeblink conditioning – as opposed to other assays of motor performance or non-cerebellar learning and memory – can be a useful tool in understanding ASD pathophysiology. As described below, we have performed new experiments suggested by the reviewers and expanded the Discussion to help make this case.

*1) Is there any cerebellar test which is not multisensory? If so, please indicate which one and then subject one or two mutants to such a task, so as to increase the impact of the association created*.

We performed new gait analysis experiments in three mouse models to evaluate motor performance that is partially influenced by the cerebellum. Importantly, this task does not require the multisensory associative learning that is modeled in delay eyeblink conditioning (although it is true that gait adjustment does require sensory feedback). As the reviewers point out, demonstration of a cerebellar, non-multisensory phenotype that is not affected by ASD-related mutations would strengthen our claim that the defects associated with cerebellar learning are particularly connected to the overall ASD phenotype. We found no significant gait differences in the three models we tested, as described (Table 1). In addition, we added information from the literature on gait and rotarod to strengthen our point (Table 2).

*2) Is there any non-cerebellar test which is not multisensory? If so, please indicate which one and then subject one or two mutants that express the relevant genes in the non-cerebellar regions involved to such a task, so as to increase the impact of the association created. This might create a double dissociation and/or association*.

We performed new swimming Y-maze acquisition experiments in three mouse models to evaluate learning and memory in a non-cerebellar task not typically related to ASD dysfunction. As the reviewers point out, demonstration that eyeblink conditioning does not simply represent a global defect in the mechanisms of learning would strengthen our claim that the defects association with cerebellear learning are particularly connected to the overall ASD phenotype. We found no significant swimming Y-maze acquisition differences in the three models we tested (Table 1). In addition, we added information on ASD-related cognitive differences in these mice (Table 2) to further strengthen our point.

*3) The deficits observed in the mice do not appear to closely resemble those observed in autistic children or adults. Sears et al. reports deficits in appropriate timing of the eyelid responses, but faster learning and larger amplitude CRs in ASD patients. Another study (Oristaglio et al.) in children with ASD also reports timing deficits. The present manuscript does not present much data on timing of CRs, save some very small changes shown in*
Figure 6*. Instead, the amplitude and probability of the CRs is smaller in some of the mice. It is not clear how to relate these differences to those observed in humans or how they fit with the well-documented role of cerebellar cortex in controlling the timing of CRs. This is a serious concern which needs to be discussed*.

We agree with the reviewers that it is important to evaluate the potential clinical relevance of our findings. We have included two new paragraphs in the Discussion on this point (“Comparison with eyeblink conditioning phenotypes in autistic persons”). We have closely analyzed the sparse literature on eyeblink conditioning in ASD patients. While the literature does point out differences between our results and the clinical results – for instance, gains in learning function in ASD patients and loss of learning function in Fragile X patients, compared with loss of learning function in our mouse models – we believe that there are important limitations to the existing clinical studies. We explore these limitations in the Discussion, and suggest ways in which our results might inform expansion of the clinical ASD eyeblink conditioning literature.

Finally, we note that the changes we have observed are quite comparable in magnitude to published findings from human subjects. In particular, differences in latency-to-CR-onset are in the same range as human literature. This is now discussed in the subsection headed “Comparison with eyeblink conditioning phenotypes in autistic persons”.

*4) Altered gene expression is global in 4 of the 5 mouse models used. Given this, it is unclear how the authors can be sure the deficits are actually due to changes in the cerebellum. Relatively small differences observed in amplitude and probability could have many origins outside of cerebellum. This is a major shortcoming which needs to be addressed, as one of the main motivations of the paper seems to be to approach the problem of understanding the link between cerebellum and ASD*.

We acknowledge that the global nature of mutations in some of our models could make our results difficult to interpret. We have added new discussion to examine how dysfunction of extracerebellar areas might account for our findings (“Extracerebellar sites”). At the same time, we also point out how cerebellar cell-type specificity in two of our models, *L7-Tsc1* and *Shank3ΔC*, strengthens the cerebellar nature of findings with respect to ASD.

*5) The phenotypes are mostly relatively subtle, but at the same time compelling in that all ASD mouse models show some deficit. To what extent is the eyeblink test specific for autism? Wouldn't all specific cerebellar tests show some phenotype in the ASD mouse models tested here? Please note that tests like rotarod are not cell specific*.

We agree with the reviewers that eyeblink conditioning requires specific justification. This is now done, and as mentioned in our response to major point #1, we now provide information about other phenotypes that have partial cerebellar involvement (gait and rotarod) to emphasize our point about eyeblink conditioning: gross motor performance does not map to brain circuitry (Figure 7), or to ASD phenotypes (Table 2), with the precision of our delay eyeblink conditioning data. Furthermore, in the Discussion (second paragraph) we explore the advantages of using eyeblink conditioning as a probe of cerebellar function.

*6) To strengthen the association between cerebellar behavioral deficits and autism, shouldn't the level of the behavioral cerebellar motor deficit be correlated quantitatively to the level of more authentic autistic behavior such as grooming, vocalization, repetitive behavior etc? If there is a positive relation between these levels, the association will be stronger. So please try to provide these correlations at a quantitative level (if necessary by digging into the literature)*.

We agree with the sentiment behind this question. However, assays of grooming, ultrasonic vocalization, and social deficits vary between labs, making it difficult to make even a rank-order comparison. Despite this difficulty, we did collect what information we could. We now supply extensive information about the ASD phenotypes from the literature, including the direction and the significance of the results (Table 2). We suggest that a qualitative point stands out: delay eyeblink conditioning is the one non-cognitive phenotypes that correlate strongly with cognitive and social deficits. This strengthens the specificity of the link between cerebellar learning and ASD function (end of Discussion, “The cerebellum in cognition and autism”).

*7) Are there mouse mutants that do show cerebellar deficits, but no signs of autism (as described above)? Please indicate and/or show which ones*.

This possibility has been explored by [41]. In their experiments, they find no such link, putting it seemingly at odds with [119], as well as our own findings. We discuss contrasts among these studies that might be resolved in the future (in the last paragraph of the subsection headed “The cerebellum in cognition and autism”).

*8) Please provide a more detailed description of the precise cellular distribution of the expression of all five genes involved within the entire olivocerebellar system as well as the non-cerebellar regions involved in eyeblink conditioning (e.g. trigeminal and facial nucleus and red nucleus) in the Introduction, Methods or Results section (a table would also be nice), so the reader can better relate the phenotypes later on to the anatomy. The authors already did quite a good job on this point, but it can be made more extensive, which in this case is relevant as the models concern mostly global mutants*.

In the Discussion, we provide a more extensive description of the involvement of extracerebellar areas in this phenotype (“Extracerebellar sites”). We also provide additional information about gene expression in these areas (Figure 7—figure supplement 1).

*9) What is the point was of measuring probability in two different ways, first as 15% of the UR and next from the integration of the non-failure distribution? The two analyses did not appear to give statistically different outcomes, and it remains unclear what was gained by the second analysis (or retaining the first analysis). After the description of the method and the report that the data are in*
Figure 3*, nothing is stated as being learned. What is the purpose of including both*?

We have revised the manuscript to focus on only one method of analysis.

*10) Results (subsection headed “General eyeblink conditioning defects”): “These findings indicate that delay eyeblink conditioning can be used to further probe defects specific to the initial acquisition of conditioning, a process that requires the cerebellum.” This concluding statement from the primary experiment of manuscript does not make sense. The first half (conditioning can be used to probe conditioning) is tautological, and the second part (cerebellar dependence) was known before the experiments were done. The sentence also doesn't follow logically from the observation that extinction was normal and acquisition was altered. Statements like this give the sense that this is an archival set of measurements that were definitely worth making and may be of value in a deeper investigative context, but that there really are no major insights gained from the work*.

We thank the reviewers for pointing this out. We have reorganized the Results section to remove such statements. We now use the Discussion section to show how the data fit into a framework of cerebellar function.

*11) It makes sense to measure both response probability and amplitude, since these may reflect different aspects of the learning, but the presentation of the data is bizarre and makes it hard to follow the point. In the subsection headed “Performance differences associated with olivocerebellar pathway perturbations”, it is almost as though another writer took over the paper. The text reverts to explaining the circuit, restates the purpose of the experiments, re-identifies the genes, and re-calls*
Figure 1*. All the data that actually matter in the manuscript are in*
Figures 4 and 5
*(and the raw traces of*
Figure 2
*are helpful as well as illustrations of data that gave rise to these summaries). The plots of*
Figures 4 and 5
*catalogue the defects associated with different genes (useful) but are not interpreted in a way that is particularly compelling. In the last paragraph of the aforementioned subsection comes the conclusion of three of the models, “Taken together, [these] models suggest that Purkinje cell disruption is accompanied by deficits in learning.” This is not a new discovery; it is a confirmation of something long known. It seems the cart is before the horse here; it is plausible that the discovery is that these autism-linked mutations alter the cerebellar circuit sufficiently to affect learning processes known to depend on the cerebellum. That the effect can be localized to Purkinje cells, however, is not convincing*.

We thank the reviewers for this comment. As mentioned in major point #10, we have reorganized the Results and Discussion sections to avoid the problem described by the reviewer.

*12) Ascribing the changes in amplitude in the* MeCP2 *and* Shank *mutants to the “mossy fiber pathway,” which means granule and Golgi cells, is not convincing. For* Shank*, given that the protein is expressed in the cerebellum only in granule cells (*Figure 1*), any changes in these cells may well contribute to deficits, but why this should affect amplitude specifically is not clearly justified. For the* MeCP2 *mutants, the more widespread expression does not make the link to the mossy fiber pathway make sense. Besides, as defined, the mossy fiber pathway is everything downstream from the mossy fibers, which encompasses the full cerebellum. It should also affect Purkinje cell activity, which the authors just said would necessarily affect response probability. A key piece of information missing is what the expression of all proteins is in mossy fibers, which are heterogeneous. Besides, without knowing what the nature of the signals made by Purkinje cells is under the different conditions, these types of conclusions cannot be drawn with certainty (this does not require redoing all the experiments with electrophysiological recordings; just being realistic in the written interpretations)*.

We now describe more extensively the possible involvement of these genes in mossy fiber-granule cell function and put this in the context of basic research on granule layer function. We discuss the limits to the interpretability of our evidence in the Discussion (in the subsection headed “Putative substrates for performance defects: the granule cell pathway”).

[Editors' note: further revisions were requested prior to acceptance, as described below.]

*The referees appreciated the effort the authors have made to address their concerns and agreed that the manuscript is much improved. However, it was felt that the authors need to go further in order to tone down the rhetoric in the discussion regarding eyelid conditioning representing a model system for multisensory learning. It is also strongly recommended that the authors drop “multisensory” from the title. Detailed suggestions for revision from the three reviewers are below*.

We thank the three reviewers and two editors for their detailed attention to our manuscript. In the revised manuscript we respond to all remaining comments. Specifically, the word “multisensory” no longer appears in the manuscript. In addition, we have rewritten the last paragraph of Discussion to get away from the multisensory concept, which we agree was too broad.

Reviewer #2:

*Though I appreciate the authors' efforts and believe that the data in this paper will be useful for future investigations into the intriguing links between cerebellum and ASD, I still have concerns about the overall significance of the findings as well as the authors’ interpretation. Several other papers (*[62]*;*
[82]*) have already documented deficits in eyelid conditioning in rodent models of autism and related disorders. These include a recent paper in Nature Communications from Piochon et al. (with some of the same authors as Kloth et al.) that documents eyelid conditioning deficits in one of the same mouse lines that is used in the present study. While it is true that the present manuscript shows that eyelid conditioning deficits are common to several different autism models, there doesn't seem to me any compelling pattern or mechanistic explanation for the various deficits that the authors observe in the various models. The facts that (1) the deficits are fairly subtle and vary across the different lines and (2) many of the disrupted genes are expressed outside the cerebellum and/or at multiple sites within the cerebellum make it impossible to link the phenotypes with known models/mechanisms of cerebellar learning. The authors’ attempts to make such links in the Discussion were not convincing*.

We disagree strongly. It was by no means a foregone conclusion that we would discover delay eyeblink conditioning deficits in every model that had ASD-like phenotypes. We have also demonstrated several types of dissociation (for which we are grateful to the reviewers for suggesting we do). We are now confident that our findings are significant and nontrivial.

*I also am still unconvinced that eyeblink conditioning is a paradigm for studying multisensory deficits in autism. The authors state in the Discussion that “Learning that an auditory or visual stimulus predicts some other sensory event is likely to be at the core of how infants gain cognitive capacities. In this context, eyeblink conditioning is unusually interesting because it captures a central component of statistical learning, the close timing of two events. Future research may identify parallels between delay eyeblink conditioning and other forms of learning that rely on the cerebellum*.*”*

*This kind of “statistical learning” is what one imagines is going on in cerebral cortex. For that matter, associating two closely spaced events could occur almost anywhere in the brain via Hebbian plasticity. One thinks about the cerebellum as a structure for supervised learning or learning from error. Virtually all real-life learning is multisensory, so I really don't see the specific connection to the cerebellum. Deficits in cerebellar function may well be playing a special/critical role in autism, as highlighted in Dr. Wang's excellent recent review. The idea of disrupted internal model learning is particularly intriguing in relation to the cerebellum-autism connection. However, I am not convinced by the multisensory learning/eyelid conditioning angle pushed here*.

We agree that the cerebral cortex is a prime suspect for engaging in statistical learning. However, to our knowledge it is unknown whether the information processing that supports such learning is performed entirely by one brain region, or distributed in some manner across brain regions. We have rewritten the last paragraph of Discussion to remove the word “multisensory” and to link the work to temporally precise associative learning. We have also removed “multisensory” from the Title and the rest of the manuscript.

*Another concern voiced in my original review related to the unclear relationship between the eyelid conditioning deficits in mice and those reported previously in individuals with ASD. I appreciate that methodological differences complicate comparisons, but the fact that there is not a close alignment of the animal and human studies remains concerning to me*.

In response to the original review, we added a thorough discussion of human eyeblink conditioning. We pointed out areas that require future experiments, and concluded that “past findings suggest that perturbation of cerebellar granule cell layer activation may be common in both idiopathic and syndromic autism, with impacts that vary by specific genotype and disease model.” We have now added text to make an explicit mouse-vs-human comparison. The text appears in the subsection headed “Comparison with eyeblink conditioning phenotypes in autistic persons”.

*In the first round of reviews one of the reviewers commented that “…statements like this give the sense that this is an archival set of measurements that were definitely worth making and may be of value in a deeper investigative context, but that there really are no major insights gained from the work.” This statement sums up my overall feeling about this paper*.

In the face of this, we thank the reviewer for his/her willingness to provide a vigorous critique.

Reviewer #3:

*1) Introduction, first paragraph. “Sensory responsiveness and social symptoms are strongly correlated in high-functioning autism patients.” It is not clear what is meant by “social symptoms” or how they would vary along an axis*.

Reworded to reflect the original Hilton et al. paper.

*2)*
Figure 1*, please define and explain the red and blue (portions of the) records. I think blue is CS only in a trained animal but it is not explicitly stated*.

The caption is now repaired.

*3) In the Results, subsection “Normal extinction and reacquisition of conditioned responses”, the authors state: “…eyeblink extinction and savings, two processes ascribed to the deep cerebellar nuclei…”. This statement is inaccurate, probably because it is written as shorthand. Extinction requires the cerebellar cortex (and the whole olivocerebellar loop) as shown in Medina Nores and Mauk 2002 Nature. It is argued that savings results from plasticity in the deep cerebellar nuclei, but re-acquisition of well-timed responses (facilitated by savings) likely requires the cerebellar cortex (*[76]
*J Neurosci). Please edit for accuracy. (Also eyeblinks are not extinguished; conditioned responses are extinguished*.*)*

We have altered the text to reflect a more correct understanding of extinction and savings.